

# Technical Note: Testing pore-water sampling, dissolved oxygen profiling and temperature monitoring for resolving dynamics in hyporheic zone geochemistry

Tamara Michaelis[1], Anja Wunderlich[1], Thomas Baumann[1], Jürgen Geist[2], and Florian Einsiedl[1]

[1]Chair of Hydrogeology, School of Engineering and Design, Technical University of Munich (TUM), Munich, Germany
[2]Chair of Aquatic Systems Biology, School of Life Sciences, Technical University of Munich (TUM), Munich, Germany

**Correspondence:** Florian Einsiedl (f.einsiedl@tum.de)

**Abstract.** The hyporheic zone (HZ) is of major importance for carbon and nutrient cycling as well as for the ecological health of stream ecosystems. However, biogeochemical observations in this ecotone are complicated by a very high spatial heterogeneity and temporal dynamics. Especially the latter are difficult to observe without disturbing the system. In this field study, we tested and combined three less common methods for time-resolved measurements with high vertical resolution. We installed Rhizon

samplers for repeated pore-water extraction, an optical sensor unit for in-situ measurements of dissolved oxygen, and a depth-resolved temperature monitoring system in the HZ of a small stream. While Rhizon samplers were found to be highly suitable for pore-water sampling of dissolved solutes, measured gas concentrations, here $CH_4$, showed a strong dependency of the pump rate during sample extraction, and an isotopic shift in gas samples became evident. This was presumably caused by a different behaviour of water and gas phase in the pore-space. The manufactured oxygen-sensor could locate the oxic-anoxic interface

with very high precision. This is ecologically important and allows to distinguish aerobic and anaerobic processes. Temperature data could not only be used to estimate vertical hyporheic exchange, but also depicted sedimentation and erosion processes. Overall, the combined approach was found to be a promising tool to acquire data for the quantification of biogeochemical processes in the HZ with high spatial and temporal resolution.

## 1 Introduction

The hyporheic zone (HZ) is the interstital habitat below streams and rivers, adjacent to and influenced by the stream water above and the groundwater below (Peralta-Maraver et al., 2018). The importance of this zone for stream ecosystems has long been recognized (Boulton et al., 1998) and is emphasized until today (Lewandowski et al., 2019). Ecosystem functions of the HZ include rapid carbon and nutrient recycling (Findlay, 1995; Sophocleous, 2002), physical, chemical, and biological filtration of streamwater (Hancock et al., 2005), and flood wave retention (Boulton et al., 1998). It also serves as a habitat for

microbiota and macrozoobenthos (Hendricks, 1993; Robertson and Wood, 2010), provides spawning grounds for fish (Malcolm et al., 2005; Sternecker and Geist, 2010; Smialek et al., 2021), and is important as a juvenile habitat for endangered freshwater mussels (Auerswald and Geist, 2018; Denic and Geist, 2015). On the other hand, as a result of the high microbial activity,



greenhouse gas (GHG) production can be substantial in the HZ (Trimmer et al., 2012; Stanley et al., 2016), making many rivers net methane ($CH_4$), nitrous oxide ($N_2O$) and carbon dioxide ($CO_2$) emitters (Romeijn et al., 2019; Saunois et al., 2020). Therefore, a deep understanding of the processes in the HZ is essential in many disciplines (Krause et al., 2011). High spatiotemporal heterogeneity is making data acquisition for model development and calibration a challenge (Braun et al., 2012). The HZ is a complex system, influenced by many interrelated factors and more observations are needed to better describe the hydrological, geochemical and ecological functioning of this dynamic zone.

Several methods have been described to investigate HZ biogeochemistry. Well known approaches are direct sediment sampling or pore-water sampling from sediment cores. Water samples can be extracted from cores by centrifugation (Emerson et al., 1980), squeezing (Bender et al., 1987), or pressurization (Jahnke, 1988). However, coring, transportation, and water extraction may disturb the sample and significantly deteriorate sample quality. Sediment sampling also disturbs the sampling site, limits spatial resolution, and can change geochemical gradients through the introduction of bypass flow along boreholes and sampling devices. These issues are critical in the HZ, where geochemical gradients are often steep. Pore-water equilibrium dialysis samplers (peepers), as first described by Hesslein (1976), can be used to obtain pore-water concentration profiles without coring at a high vertical resolution (e.g. Michaelis et al., 2022). A disadvantage is that samples represent an average over the sampling period of (usually) several weeks, making it impossible to observe short-term temporal dynamics typical for the HZ (Boano et al., 2014). Further, both sampling from sediment cores or peepers is not suitable for long-term observations due to perturbation during sampling and the necessity to sample at slightly different positions. Further, air contamination during sample extraction from sediment cores or peeper chambers is likely which is a problem when studying anoxic processes. For in-situ measurements, microsensors have been developed which can be driven into the sediment to record dissolved $O_2$ or $HS^-$ concentrations, pH and redox potential with a vertical resolution in the mm range (Boetius and Wenzhöfer, 2009). These sensors have been employed at the sea floor (e. g. Vonnahme et al., 2020), but they are not suitable for rivers or streams with high flow velocities or coarse-grained sediments due to their high fragility. In addition, sensors and additional instrumentation for precise handling are very expensive.

This leaves the observation of short- and long term temporal fluctuations in HZ biogeochemistry a challenge. In this study, we were looking for an innovative set of techniques for constraining these temporal dynamics. The aims were: first, to identify a technique of non-invasive pore-water extraction with a high temporal and vertical resolution for point-observations of water chemistry; second, to get reliable, high-resolution dissolved-oxygen gradients to distinguish oxic and anoxic conditions; and third, to find a way to monitor system changes between sampling campaigns for a better interpretation of geochemical data.

We tested and combined three previously described but less known low-cost methods, focusing on temporal changes in hyporheic $CH_4$ cycling: As a first component, 15 Rhizon samplers (microfilter tubes) were installed for repeated pore-water sampling. Rhizon samplers, typically applied for soil moisture measurements in the unsaturated zone, have sporadically been used for pore-water extraction: Shotbolt (2010) used Rhizon samplers for pore-water extraction from sediment cores, Seeberg-Elverfeldt et al. (2005) in combination with an in-situ chamber in the Wadden sea, and Song et al. (2003) to sample pore-water from lake sediment microcosms. The second component was a custom-coated fiber-optical oxygen sensor unit based on the description of Brandt et al. (2017) for a precise allocation of the oxic-anoxic interface. As a third component, temperature





monitoring in 14 different depths was used for an estimation of hyporheic exchange. Flux rates were calculated with analytical models introduced by Hatch et al. (2006) and Keery et al. (2007) using the software package VFLUX (Gordon et al., 2012).

We hypothesize that this combined approach would be suitable for a high-resolution spatiotemporal HZ monitoring to resolve changes in the geochemistry, particularly the methanogenic and methane consuming zones, for short events (e.g. during and after a flood event) and in the long run (e.g. seasonal variations).

## 2 Methods

### 2.1 Study site and station design

The study was conducted at the Moosach river in southern Germany, close to the city of Freising. The river is characterized by a low gradient, a high fraction of fines and stable hydrologic conditions (Auerswald and Geist, 2018). The sampling station was installed at the right bank of the river in a low-flow zone with fine, organic-rich deposits. The grain size distribution of the deposits consisted of 3 % gravel, 27 % sand, and 70 % silt with a porosity of 81.5 % (see App. A). The organic matter content was 21 %. High $CH_4$ production was expected due to the high content of fines and organic matter (Bodmer et al., 2020). Water

depth at the site was approximately 0.6 m. After installation, we observed heavy sedimentation and during the summer months, mainly between July and September, major macrophyte growth.

The sampling station comprised 15 Rhizon samplers for depth-resolved pore-water sampling (Sec. 2.2), a self manufactured oxygen sensor (Sec. 2.5), and 14 temperature sensors (Sec. 2.6). Fig. 1 shows all components of the sampling station. Rhizon samplers and temperature sensors were fixed horizontally on opposite sides of a PMMA (Plexiglas) panel. The panel was

inserted longitudinally to the flow direction in order to keep disturbances to river flow and horizontal hyporheic fluxes to a minimum. Rhizon samplers were facing towards the main channel while temperature sensors were facing towards the river bank. A swimming raft allowed access to the tubes connected to the Rhizon samplers to guarantee sampling without sediment disruption. Temperature sensors were connected to data loggers installed on land next to the river. A fiberoptical measurement system for oxygen concentration was placed right next to the sampling station. With the custom made optical sensor, an oxygen

meter and an optical fiber, $O_2$ saturation could be measured with a depth-resolution of 1 cm.

### 2.1.1 Pore-water sampling with Rhizon samplers

Our sampling station was equipped with 15 Rhizon samplers with a pore diameter of 0.12-0.18 $\mu m$ and a filter length of 5 cm (Rhizosphere, Wageningen, The Netherlands). The samplers were fixed horizontally with 3 cm distances. Polytetrafluorethylene (PTFE) tubes with 1.32 mm inner diameter (Cole Parmer, St. Neots, UK) were connected to the samplers to lead

pore-water samples to the water surface. The material was chosen for its low gas permeability.

Samples were withdrawn simultaneously from all 15 Rhizon samplers with two ISM 1089 Ismatec Ecoline Peristaltic pumps (VWR International, Darmstadt, Germany) with eight cassettes each and gastight Viton peristaltic tubing with an inner diameter of 0.51 mm (Cole-Parmer GmbH, Wertheim, Germany). Three pump rates between 0.01 mL min$^{-1}$ and 0.38 mL min$^{-1}$ were





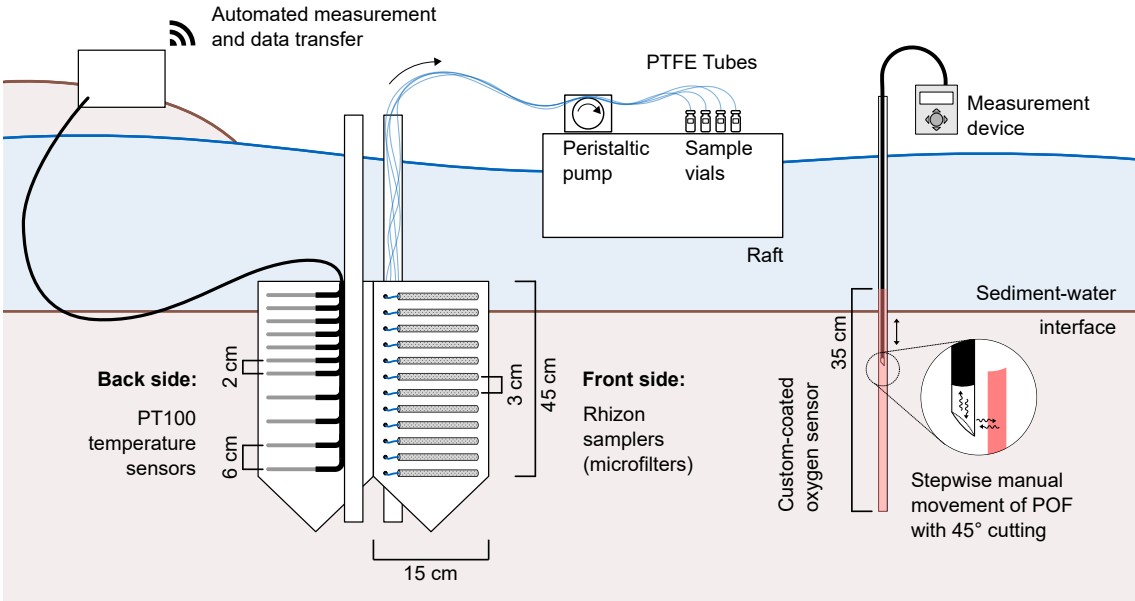

**Figure 1. Design of the monitoring station at River Moosach, Freising, Germany.** For reasons of clarity, the schematic figure does not show all sensors.

tested. It should be mentioned that the application of a vacuum results in degassing. As the actual pressure conditions can not
be measured, this change of the sample cannot be fully quantified. Calculations indicate that the effect is more pronounced at
higher gas concentrations and affects not only the gases but also the pH-value and the concentration of bicarbonate.

Samples for stable water isotopes, anion- and cation analyses were collected in 1.5 mL glass vials without headspace. For
gas analyses, 10 mL glass vials were crimped gastight with butyl rubber stoppers and flushed with synthetic air ($O_2$, $N_2$).
3 mL synthetic air were removed from the enclosed vials right before sampling. Rubber stoppers were then pierced with
95  needles connected to the peristaltic tubing and 3 mL of sample were pumped directly into the vial, providing a completely
gastight, pressure compensated sampling technique. Samples for gas analyses were fixated with 20 µL 10 M NaOH (Carl Roth,
Karlsruhe, Germany). For sulfide measurements, 15 mL Falcon tubes were prepared with 1 mL 1 M zinc acetate (Carl Roth,
Karlsruhe, Germany). A sample of 4 mL was injected slowly from below to allow precipitation of ZnS before air contact. All
samples were transported in a cooler and stored refrigerated prior to analysis.

### 2.1.2   Pore-water sampling with a peeper

100

As second pore-water sampling method, a pore-water dialysis sampler (peeper) was used. The body of the peeper was equipped
with 2 columns of 38 chambers, each being filled with deionized water and covered with a semi-permeable membrane (pore
diameter 0.2 µm) (Pall Corporation, Dreieich, Germany). Over a period of 3 weeks, an equilibrium between the water in the
chambers and the surrounding pore-water was obtained. Immediately after removing the peeper from the sediment, the water





from the chambers was withdrawn with syringes and injected into vials. Due to the low amount of available sample volume (on average 3 mL per chamber), pore-water analysis was restricted to anion, cation and $CH_4$ concentrations along with the stable carbon isotope ratio ($\delta^{13}C$) of $CH_4$. Samples for anion and cation analysis were stored in 1.5 mL glass vials. Samples were fixated with $10\,\mu L$ 0.5 M NaOH (anions) and $10\,\mu L$ 1 M HCl (cations) to cope with long analysis times due to the large number of samples. Vial preparation for gas analyses, including fixation, flushing and sealing, was similar to the sampling method described in Sect. 2.1.1. During sample injection, two syringes were used, one for the sample and one to allow pressure exchange. Both needles were removed directly after sampling.

Dissolved $O_2$ concentrations were measured in the field immediately after retrieval of the peeper from the sediment and its cleaning with de-ionized water. A Clark-type microsensor (Unisense, Aarhus, Denmark) was pierced through the membrane for the measurements (Revsbech, 1989). A time constraint to this technique is contamination with atmospheric $O_2$ which can diffuse quickly through the membrane under air contact. Thus, $O_2$ measurements had to be conducted as rapidly as possible and only selected chambers were tested to avoid artefacts.

## 2.2 Analytical methods for pore-water analysis

Anion and cation concentrations were measured with a system of two ICS-1100 ion chromatographs (Thermo Fisher Scientific) equipped with Dionex IonPacTM AS9-HC and CS12A columns, respectively. All results represent an average of triplicate measurements and were evaluated based on seven calibration standards (Merck, Darmstadt, Germany) reaching an analytical uncertainty of $< 10\,\%$. Detection limits were $0.039\,\mathrm{mmol\,L^{-1}}$ for $Ca^{2+}$, $0.032\,\mathrm{mmol\,L^{-1}}$ for $Mg^{2+}$, $0.020\,\mathrm{mmol\,L^{-1}}$ for $Cl^-$, $0.012\,\mathrm{mmol\,L^{-1}}$ for $NO_3^-$, $0.007\,\mathrm{mmol\,L^{-1}}$ for $NO_2^-$, and $0.008\,\mathrm{mmol\,L^{-1}}$ for $SO_4^{2-}$.

Stable water isotopes were measured in the same vials which had been used for cation analysis or in completely filled 1.5 mL glass vials that had been sampled separately. Only samples without acid or base addition for fixation could be used. Fixation was necessary for peeper samples and Rhizon samples for the median pump rate of $0.19\,\mathrm{mL\,min^{-1}}$ (same sampling date) due to the high number of samples and long expected analysis times. Samples were analyzed with the IWA-45EP isotopic water analyzer (Los Gatos Research, San Jose, USA) calibrated with 3 standards (USGS Reston Stable Isotope Laboratory, Reston, USA) with an analytical error of $< 0.1\,\%o$ for $\delta^{18}O$ and $< 1\,\%o$ for $\delta^2H$. Results are expressed in the $\delta$ notation relative to the V-SMOW standard. Deuterium excess was calculated as $d = \delta^2H - 8 \cdot \delta^{18}O$ (Dansgaard, 1964).

Methane concentrations were measured according to a procedure introduced by the US Environmental Protection Agency EPA (2001) adopted to small sample volumes. Before analysis, vials were left for equilibration at $30\,°C$ for at least 2 hrs. Headspace $CH_4$ concentrations were measured with a Trace 1300 gas chromatograph (GC) (Thermo Fisher Scientific, Dreieich, Germany) with a TG-5MS column and flame-ionization detector (FID), calibrated with 3 concentration standards (Rießner Gase, Lichtenfels, Germany). Samples were measured in triplicates of $250\,\mu L$ manual headspace gas injection. Calculations of total concentrations before equilibration with the headspace were based on Henry's law as previously described (Kampbell and Vandegrift, 1998; EPA, 2001).

The vials for $CH_4$ concentration measurements were also used for isotopic analyses with a G2201-i gas analyzer (Picarro, Santa Clara, USA) for $^{12}C/^{13}C$ ratios in $CH_4$ with an analytical uncertainty of $< 0.16\,\%o$. Headspace vials were directly connected to





the Small Sample Introduction Module (SSIM) with needles. Dilution of the samples with synthetic air and re-pressurization

of the glass vials was necessary for repeated measurements due to the small sample- and headspace volume. Reliable results could not be obtained at headspace $CH_4$ concentrations of $< 30\,ppm$ (Michaelis et al., 2022). Results are represented in the $\delta$ notation relative to the V-PDB standard.

Sulfide samples were reactivated in the laboratory by adding $50\,\mu L$ $49\,\%$ $H_2SO_4$ to dissolve the ZnS precipitate directly before analysis with the 1.14779.001 Spectroquant Sulfide Test for the Spectroquant Prove 100 photometer (Merck, Darmstadt, Ger-

many). Sulfide concentrations were found to be below the detection limit of $0.02\,mg\,L^{-1}$ during several sampling campaigns and were therefore excluded from subsequent sampling and analyses. This may be indicative of very low sulfide concentrations in the HZ, but an issue with sampling or analytical methods cannot be ruled out.

## 2.3 Statistical analyses

$CH_4$ concentration, $\delta^{13}C$-$CH_4$, $\delta^{18}O$-$H_2O$, $\delta^2H$-$H_2O$, $Ca^{2+}$, $Mg^{2+}$, and $Cl^-$ concentration data from peeper and Rhizon

measurements at different pump rates were tested for statistically significant differences. First, data sets were checked for normal distribution with the Shapiro Wilk test and a visual inspection of box plots. Levene's test was used for assessing the homogeneity of variance. Since the requirements for t-tests and the one-directional ANOVA test (normal distribution of all data sets and for ANOVA, homogeneity of variances) were not met for all data sets, nonparameteric tests were chosen. The Mann Whitney U test was applied for pairwise comparisons and the Kruskall Wallis H test for assessing differences in more than

two data-sets, comparing all sampling techniques for each parameter. In addition, independent t-tests were used for pairwise comparisons where both data sets were normally distributed. All assessments were implemented in python (version 3.8.3) using the scipy.stats package (version 1.5.1).

## 2.4 Dissolved oxygen profiling

Measuring $O_2$ concentrations in extracted samples had two major disadvantages: sample contamination with atmospheric $O_2$

during extraction could not be securely excluded and the vertical resolution of $3\,cm$ between the Rhizon samplers was too low to depict the steep $O_2$ gradient. Therefore, a system for in-situ oxygen profiling was constructed and installed.

Following the example of Brandt et al. (2017), an optode for optical $O_2$ measurements was manufactured by coating a Plexiglas tube with an oxygen-sensitive dye. To produce the sensing element, a sensor cocktail was prepared by dissolving $20\,mg$ of platinum tetrakis(pentafluorophenyl)porphyrin (PtTFPP) (Porphyrin Systems, Lübeck, Germany) and $2\,g$ polystyrene in $10\,mL$

toluene. The sensor cocktail was filled into a glass tube with a punched Viton septum (diameter $4.5\,mm$) at the lower end where the PMMA tube with an outer diameter of $5\,mm$ (inner diameter of $3\,mm$) fits tightly. The PMMA tube was then pulled through the sensor solution with a stepper motor at $0.25\,cm\,s^{-1}$ and left to dry for at least $12\,hrs$ yielding a thin oxygen-sensitive coating on the outside of the tube. Measurements were performed with the Fibox 4 Trace Oxygen Meter (PreSens, Regensburg, Germany) connected to a polymeric optical fiber (POF) with an outer diameter of $2.7\,mm$. The tip of the POF was

equipped with a $45\,°$ cutting to allow signal transfer orthogonal to the fiber (see Fig 1).





In contrast to the work of Brandt et al. (2017), the sensor was not connected to an automated motor unit for data recording due to the low stability of the long Plexiglas tube ($> 75$ cm above the sediment-water interface at a water depth of 60 cm) and the risk of water-level changes at high flow. Instead, measurements were performed manually by pulling up the POF in 1 cm steps as marked on the cable. At each depth, at least 3 measurements were done at a rate of 1 Hz. For each depth, mean and standard
deviation of repeated measurements were calculated.

For calibration, distilled water with seven different $O_2$ concentrations was prepared by stripping with $N_2$ or He gas for different amounts of time. Each sensor was installed in a flow-through cell which was flushed with the de-oxygenated water. Dissolved $O_2$ concentration in the flow-through cell was in parallel measured with a microsensor (Unisense, Aarhus, Denmark). For temperature control, the flow-through cell was placed in a column connected to a WCR-P22 thermo-controlled water bath
(Witeg, Wertheim, Germany). Calibration was conducted at 20 °C. For each sensor, temperature dependence at 0 % and 100 % air saturation (a. s.) was evaluated with 5 and 4 temperatures between 5 °C and 30 °C, respectively. Details on calibration results and calculation of dissolved $O_2$ concentrations from measured phase angles can be found in App. B.

### 2.5 Vertical hyporheic exchange estimation using temperature measurements

Temperature was measured in 14 different depths to trace hyporheic exchange fluxes at the sampling site. The four-wire PT 100
sensors (Omega Engineering, Norwalk, USA) with an accuracy of $\pm$ 0.03 °C were calibrated in a WCR-P22 water bath (Witeg, Wertheim, Germany) with an accuracy of $\pm$ 0.1 °C at seven different temperatures between 0 °C and 30 °C before installation in the field. During calibration, sensor recordings were compared to the average temperature considering all sensors yielding a constant correction factor for each sensor.

Onsite, the sensors were installed with a 2 cm depth-resolution for the first 15 cm and a 6 cm resolution below. Another sensor
was placed approximately 20 cm below the water surface in the water column. The sensors were fixed on the back side (facing the river bank) of the panel holding the Rhizon samplers. The 14 sensors were connected to four PT104A Loggers (Omega Engineering, Deckenpfronn, Germany) and a Raspberry Pi based control unit for automated data acquisition every 5 min.

Due to the long installation time, four out of 14 sensors stopped functioning properly, two additional sensors were excluded from analysis due to data gaps of $> 24$ hrs. Data processing included removal of outliers $< 0$ °C or $> 30$ °C, interpolation over
data gaps $< 24$ hrs, and re-sampling to equally spaced 5 min intervals.

Vertical hyporheic exchange rates were estimated using the software package VFLUX (Gordon et al., 2012). The software implements analytical solutions (Hatch et al., 2006; Keery et al., 2007) to the one-dimensional heat transfer equation for steady fluid flow through a homogeneous porous medium (Stallman, 1965). These solutions use amplitude and phase change in the sinusoidal diurnal signal of a pair of two temperature sensors in different depths for the calculation of the advective
flow component. VFLUX first obtains the diurnal oscillation signal by filtering the data using dynamic harmonic regression (DHR) (Young et al., 1999). Then, differences in amplitude and phase are extracted for each periodic cycle. The software calculates vertical flux rates for each specified sensor pair in $m\,s^{-1}$ based on both amplitude and phase change for each of the methods described by Hatch et al. (2006) and Keery et al. (2007). Sediment-specific input parameters for the calculations are summarized in Tab. 1.





**Table 1.** Parameters for vertical hyporheic exchange estimation using the software package VFLUX.

| Parameter | Description | Value | Source |
|---|---|---|---|
| $n$ | Total porosity | 81.5 % | Measurements (App. A) |
| $\beta$ | Thermal dispersivity | 0.001 m | Hatch et al. (2006) |
| $\lambda$ | Thermal conductivity | $0.60 \, \mathrm{W \, m^{-1} \, K^{-1}}$ | Measurements (App. A); Dalla Santa et al. (2020) |
| $c_s$ | Volumetric heat capacity of the sediment | $0.55 \, \mathrm{MJ \, m^{-3} \, K^{-1}}$ | Dalla Santa et al. (2020) |
| $c_w$ | Volumetric heat capacity of water | $4.18 \, \mathrm{MJ \, m^{-3} \, K^{-1}}$ | Gordon et al. (2012) |

## 3 Results

### 3.1 Comparison of pore-water sampling techniques

Geochemical profiles measured in pore-water samples from peeper and Rhizon samplers showed high agreement, especially for stable water isotopes and ions. Figure 2 shows depth-profiles measured with a peeper and the Rhizon samplers at 3 different pump rates. The equilibration period of the peeper was between April and May 2022. Rhizon sampling at different pump rates was conducted in May. $NO_3^-$ and $SO_4^{2-}$ concentrations were very similar for all profiles showing steep gradients in close proximity to the sediment-water interface. The low number of samples above the detection limit, together with the steep geochemical gradients, was not sufficient for statistical evaluation. $Ca^{2+}$, $Mg^{2+}$ and $Cl^-$ concentrations were on average five to seven percent lower in the peeper data compared to Rhizon samples, but different pump rates did not have an effect on average concentrations (App. C).

Average $CH_4$ concentrations in Rhizon samples deviated by -30 % (lowest pump rate) to +100 % (highest pump rate) from peeper samples. While the $CH_4$ concentration profiles recorded with the peeper showed a smooth gradient, profiles from Rhizon measurements showed large concentration differences in consecutive depths. Average measured concentrations were significantly different not only between peeper and Rhizon samples, but also for different pump rates (Fig. C3).

To analyze if isotope fractionation processes influence the measurements of dissolved solutes and gases, stable water isotopes ($\delta^{18}O$ and $\delta^2H$) were measured in water samples and stable carbon isotopes ($\delta^{13}C$) in methane. Water isotopes were only measured at the highest and lowest pump rate. Results were found to be similar, neither the t-test nor the Mann Whitney U test showed significant differences (App. C). Table 2 shows water isotopes from pore-water samples and surface water samples. Deuterium excess in the sediment was 0.5 ‰ higher in pore-water compared to surface water samples. This is below the analytical precision for $\delta^2H$ measurements of 1 ‰. $CH_4$ had a lighter isotopic composition in peeper samples compared to samples extracted with Rhizon samplers. The stable carbon isotopic composition of $CH_4$ was most heavy at the lowest pump rate. Variance in both $CH_4$ concentration and stable isotope measurements increased with increasing pump rate.





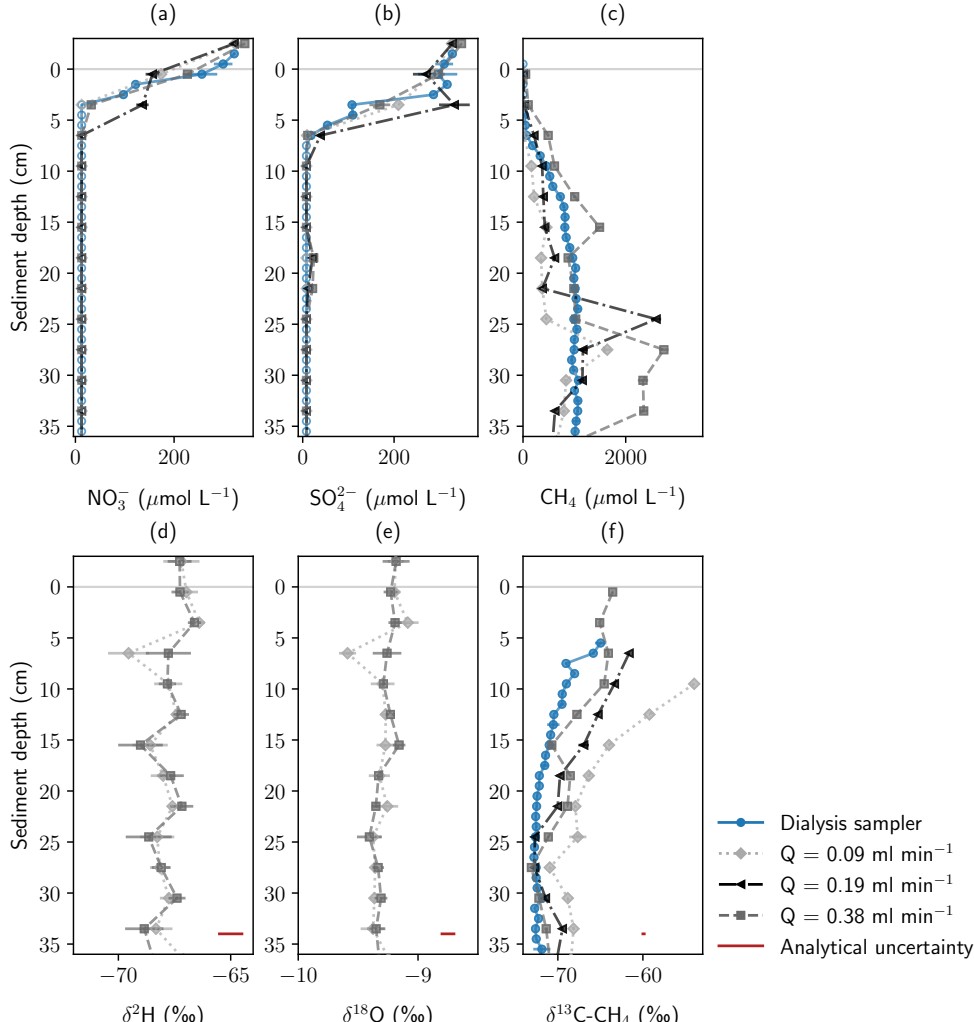

**Figure 2. Concentration and stable isotope profiles measured with a pore-water dialysis sampler and Rhizon samplers from the monitoring station at three different pump rates.** All samples were withdrawn in May 2022. Panels show (a) $NO_3^-$, (b) $SO_4^{2-}$, and (c) $CH_4$ concentration, (d) and (e) stable water isotopes, and (f) stable carbon isotopes in $CH_4$. Error bars show standard deviation of repeated measurements. In addition, analytical uncertainty of the measurement device is shown for isotope data.

## 3.2 Locating the oxic-anoxic interface

The fiber-optical sensor unit based on the description of Brandt et al. (2017) was tested against a microsensor in the chambers of the peeper (Fig. 3). The fiberoptical system was able to locate the oxic-anoxic interface precisely. All three repeated mea-

230  surements showed good agreement at a high resolution of 1 cm. However, the lowest $O_2$ concentration ($20\,\mu mol\,L^{-1}$) measured with the microsensor was higher than dissolved $O_2$ concentrations observed with the fiber-optical system below the oxic-anoxic





**Table 2.** Stable water isotopes ($\delta^2$H & $\delta^{18}$O) and deuterium excess d in pore-water, and surface water.

| Sample type | Date | Pump rate | $\delta^{18}$O | $\delta^2$H | d |
|---|---|---|---|---|---|
| Pore-water average | $30^{th}$ May 2022 | 0.09 mL min$^{-1}$ | -9.296 ‰ | -67.658 ‰ | 6.710 ‰ |
| | $31^{st}$ May 2022 | 0.38 mL min$^{-1}$ | -9.282 ‰ | -67.555 ‰ | 6.701 ‰ |
| Surface Water | $30^{th}$ May 2022 | | -9.186 ‰ | -67.196 ‰ | 6.292 ‰ |
| | $31^{st}$ May 2022 | | -9.183 ‰ | -67.273 ‰ | 6.191 ‰ |

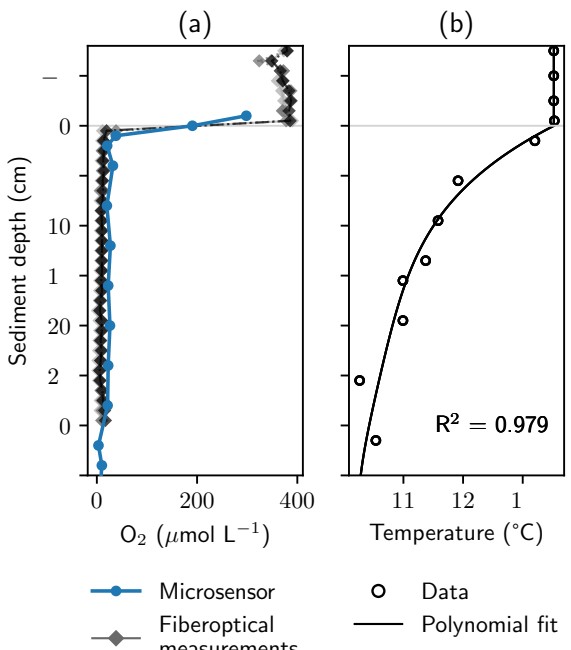

**Figure 3. Oxygen and temperature gradients at the study site.** Panel (a) shows dissolved $O_2$ profiles measured with a microsensor in the chambers of a peeper and with a manufactured in-situ fiber-optical sensor. Saturated values measured with the fiber-optical system were normalized to avoid unrealistically high values. Panel (b) shows temperature measurements and a fourth order polynomial fit which was used to calculate $O_2$ concentrations from measured phase angles.

interface. In $O_2$ saturated conditions, absolute values for calculated $O_2$ concentrations from the fiber-optical system showed high variance. Due to the flat shape of the calibration model in near-saturated conditions (see App. B, Fig. B1), signal noise led to larger errors than in the anoxic zone. Oversaturated values were normalized to avoid unrealistically high values (Eq. B4).



### 3.3 Assessing vertical hyporheic exchange

Temperature data were continuously recorded between April and August 2022. Pronounced amplitude dampening and time lag of the diurnal signal could be extracted with DHR and subsequently used for flux calculations (Fig. 4). Six sensors had to be excluded from the data set due to low data quality or larger data gaps, leaving a total of 8 sensors for the evaluation. Sensor pairs for flux calculation were chosen not to be neighbouring, but every other sensor, for example sensor 1 and 3; sensor 2 and 4; sensor 3 and 5 etc. Here, results based on the amplitude method described by Hatch et al. (2006) with the parameters from Tab. 1 are shown. Fluxes simulated with the phase method and with analytical solutions derived by Keery et al. (2007) are discussed in App. D.

Flux rates calculated with the upper three sensors showed peaks of a downward flux of up to $1 \cdot 10^{-5}\,\mathrm{m\,s^{-1}}$ ($85\,\mathrm{cm\,d^{-1}}$) in April and May 2022. Flux rates calculated between the lower five sensors showed mainly upwards directed flow. Average flux rates in 10 cm, 12 cm, and 18 cm depth were $-1.6 \cdot 10^{-7}\,\mathrm{m\,s^{-1}}$ ($-1.4\,\mathrm{cm\,d^{-1}}$), $-2.6 \cdot 10^{-7}\,\mathrm{m\,s^{-1}}$ ($-2.2\,\mathrm{cm\,d^{-1}}$) and $-4.9 \cdot 10^{-7}\,\mathrm{m\,s^{-1}}$ ($-4.2\,\mathrm{cm\,d^{-1}}$), respectively. This is shown in detail in App. D, Fig. D1, where fluxes calculated for 3 cm and 6 cm depth were excluded from the plot. Based on these values, mean water transit times in the 40 cm stretch from the bottom to the top of the geochemical profiles would be between 2 and 8 hrs.

## 4 Discussion

Our results showed good agreement for ion concentration and stable water isotope measurements in pore-water samples for the two different methods used, and equally good agreement for different pump rates when using Rhizon samplers and peristaltic pumps. This indicates high suitability of Rhizon samplers for repeated pore-water extraction at one specific site to study temporal dynamics in nutrient cycling. Certainly, Rhizons could also be used to trace the fate of contaminants, as long as the pore-diameter of the filter allows the contaminant molecule to pass and the contaminant is fully dissolved in water.

For concentration- and isotope analyses of dissolved gases, here $CH_4$, we found a lower agreement between pore-water samples extracted by Rhizons and peepers. Gas concentrations and variance increased with increasing pump rates when using Rhizon samplers. On average, concentrations were lower compared to dialysis measurements. This effect might be caused by differing behaviours of water and gas phases in the interstitial pore space. Gas bubbles might get trapped in front of the microfilters at low pump rates, because low negative pressures might not be sufficient for extraction of gas bubbles from the sediment. At higher pump rates, bubbles seem to get mobilized from a larger distance, potentially further away than liquid pore-water samples. Additionally, higher pump rates lead to a greater negative pressures which may cause increased out-gassing and thus, creation of additional gas bubbles. Since the tubes were directly connected to the sampling vials, bubbles were not lost, but gas and water phase were both contained in the sample vial. This could explain the large scatter and high concentration peaks observed at higher pump rates.

The dependence of $CH_4$ concentrations on the pump rate complicates data interpretation, because it is unknown from which part of the pore-space gas and water phase were extracted and it is difficult to define a "correct" pump rate where gas and water phase are extracted from the same pore space. One also has to consider the trade-off between low pump rates (low pressure



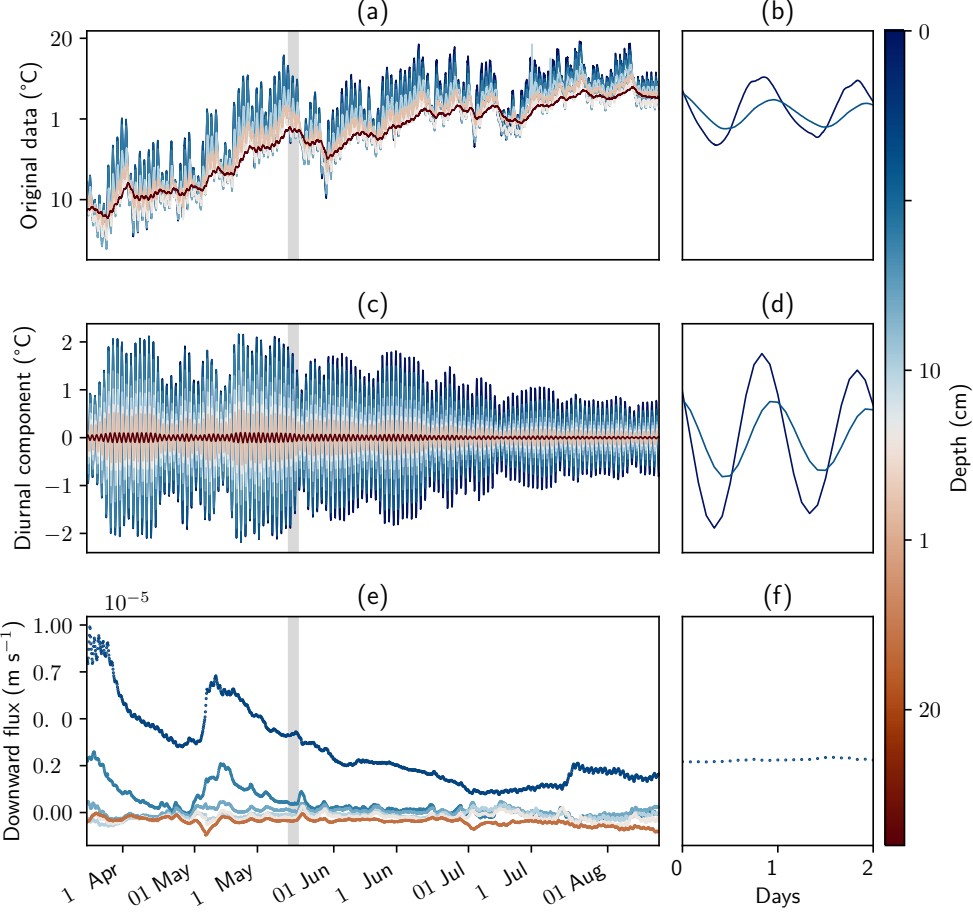

**Figure 4. Temperature measurements, filtered data and calculated fluxes.** Panels (a), (c), and (e) show the complete measurement period and all sensors. Panels (b), (d) and (f) show sensors in the surface water and 10 cm depth for a time window of two days. Panels (a) and (b) show original data. Filtered data and fluxes were calculated with the software package VFLUX and the amplitude method described by Hatch et al. (2006) using the parameters from Tab. 1.

gradient, little degassing) and corresponding sampling times (contact with air, sampling artefacts). Thus, gas measurements in pore-water samples extracted with Rhizon samplers are bound to have significant bias, especially if gas bubbles are present in

the system.

Yet, dialysis does not include the gas phase in pore-water measurements at all and it is questionable if it represents $CH_4$ distribution accurately. Bubbles can't enter the chambers of the peeper and therefore, cannot be directly sampled. Contact with the gas bubbles over extended time periods might however increase dissolved $CH_4$ concentration in the water sample. An effect could be a smoothed concentration gradient with slightly elevated concentrations. In addition, peepers integrate over several





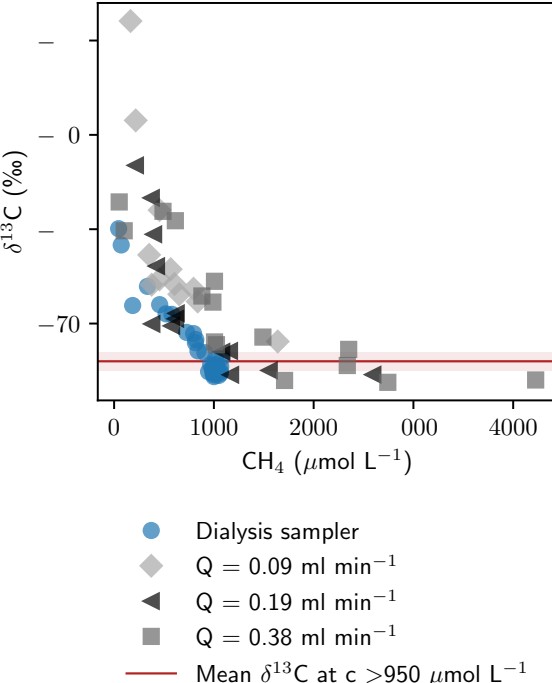

**Figure 5. Relation of CH$_4$ concentrations and isotopic composition.**

weeks while direct pore-water extraction by Rhizon samplers can capture a specific moment in time. Hence, dialysis may not
be a better solution for representing the distribution of gaseous and dissolved CH$_4$ in the sediment.

While sampling had a negligible effect on isotope fractionation for stable water isotopes, measured as proxies for the liquid
phase, $\delta^{13}$C values of CH$_4$ showed significant differences in the four measured profiles, showing an isotope fractionation
towards heavier carbon isotopes at low pump rates. At high concentrations ($> 950 \, \mu$mol L$^{-1}$), $\delta^{13}$C of CH$_4$ was found to be
similar for sampling with Rhizon samplers and peepers (-72.0 $\pm$ 1.1 ‰). Below 950 $\mu$mol L$^{-1}$, a steep non-linear increase
in $\delta^{13}$C was observed with decreasing CH$_4$ concentrations (Fig. 5). The higher stable carbon isotope composition at low
concentrations can either be caused by microbial CH$_4$ degradation (Whiticar and Faber, 1986) or by an isotope fractionation
effect during sampling, for example due to diffusion through the tubes or losses at the peristaltic pump. CH$_4$ escaping through
leakages or diffusion would lead to a greater loss of the lighter $^{12}$CH$_4$ compared to $^{13}$CH$_4$, and an enriched remaining CH$_4$
pool (Li et al., 2022). This effect is expected to be more pronounced at low concentrations. Effects of microbial degradation
would be expected to be in a similar range for peeper and Rhizon-derived profiles, thus $\delta^{13}$C values exceeding maximum $\delta^{13}$C
in peeper samples by up to 10 ‰$o$ imply fractionation during sample extraction.



Altogether, our data show that the use of Rhizon samplers for pore-water extraction has to be assessed critically for each application, mainly when working in fine sediments and considering measurement of gaseous components. Advantages are

the possibility for time-resolved measurements and that the effects and distribution of gas bubbles in the pore-space become visible. Disadvantages comprise isotopic fractionation of gaseous compounds during sampling, deviating effect of pumping on water- and gas phase, a potential underestimation of gas concentrations, and the difficulty to set optimal sampling parameters such as the pump rate.

This is true for a very fine-grained sampling site with a high content of organic matter and the occurrence of gas bubbles. In

this type of system, the extraction of pore-water requires high negative pressures at the interface between sampler and saturated sediment to overcome capillary forces in the sediment. The predominance of gas in the pore space complicates the sampling procedure and data interpretation. In sandy or gravelly river beds, lower suction rates are sufficient for pore-water extraction and $CH_4$ is likely to be present at lower concentrations, and thus, probably completely dissolved in the water phase. In these systems, the problems observed here may not be of relevance. Nevertheless, we find it important to emphasize the potential

problems of using Rhizons for gas sampling, because this has not been addressed previously in the literature and because Rhizons might get increasingly used in the future, when the interest in the HZ as an important source of GHGs rises.

Automated temperature measurements were found to be helpful in the interpretation of geochemical profiles. Temperature data can be used to characterize a site as up- or downwelling region and detect direction and dynamics in hyporheic exchange. In addition, sedimentation and erosion processes become visible in the data. The measurements can further help to improve

geochemical transport models if applied, because diffusion coefficients are temperature dependent. The installation of the sensors must be done carefully to ensure a long service life. At our field site, several sensors stopped functioning properly, most likely due to problems at soldered joints and connectors, or due to humidity and water intrusion. The software package VFLUX (Gordon et al., 2012) facilitates the use of temperature as a natural tracer for vertical hyporheic exchange by relieving authors of the non-tivial tasks of time series decomposition with signal processing techniques and implementation of analytical

models in a programming language.

The combination of pore-water sampling, in-situ oxygen profiling and temperature monitoring allowed a precise characterization of the functioning of the HZ with high spatio-temporal resolution and the three methods were found to complement each other very well. The combination could, for example, be very useful for studying the effect of floods and droughts on stream ecosystems in terms of nutrient cycling and GHG emission pulses. So far, as to our knowledge, the effect of drying

and first flush events on riverine GHG emissions has not been studied, and the described set-up would be well suited to trace the hydrological and geochemical changes in the HZ during such events. The set-up could also be used for tracer experiments, since Rhizon samplers can not only be used for pore-water extraction, but also for water injection. This could, for example, benefit the understanding of hyporheic flow patterns or the calculation of mean residence times and carbon or nutrient turnover rates. All three components could also be useful on their own or in combination with other techniques for HZ investigation.

Rhizon samplers are a low-cost option for repeated pore-water sampling, mainly suited for the study of dissolved nutrients or contaminants. Fiber-optical $O_2$ sensors present an opportunity for non-invasive dissolved $O_2$ monitoring in small streams and could supplement many existing measurement techniques in the HZ. The custom-coated sensor (Brandt et al., 2017) is a cheap



alternative to the expensive sensors available commercially. Several examples have already shown the usefulness of temperature as a tracer for hyporheic exchange (e.g. Schmidt et al. (2014); Constantz (2008); Briggs et al. (2012)), and a combination
with methods for assessing HZ geochemistry could make it an even more powerful tool.

## 5   Conclusions

In this study, we tested three methods for resolving temporal dynamics in HZ geochemistry. Rhizon samplers were found to be suitable for the extraction of water samples and measurement of dissolved solutes with a high vertical resolution. However, suitability for gas analyses was reduced, as indicated by a dependency of $CH_4$ concentration on the pump rate and a frac-
tionation towards heavier isotopes during sampling. This finding might be most pronounced in fine-grained systems with gas inclusions in the sediment, and sampling with Rhizon samplers for gas analyses might be more suitable for rivers with coarser bed substrate with higher hydraulic conductivity, where the gas is expected to be completely dissolved in the water phase. A fiber-optical $O_2$ sensor was manufactured, calibrated and tested in combination with the monitoring station. Although absolute $O_2$ concentrations in saturated and near-saturated conditions could only be determined with relatively high uncertainty, the
system was very well suited for precisely locating the oxic-anoxic interface. This parameter is highly relevant for aquatic ecology and the sensor has proven a useful, low-cost solution for HZ monitoring. The station was complemented with temperature sensors which could be used to detect sediment dynamics and estimate hyporheic fluxes. Combining the three methods has several advantages over sampling pore-water alone. Knowledge of the exact location of the oxic-anoxic interface and data on temperature and sediment dynamics between point-samplings enable better interpretation of geochemical profiles and deeper
insights into the dynamics of HZ geochemistry.

*Author contributions.* TM, AW, TB and FE conceptualized the project. TM and AW developed the methodology. TM was responsible for field work, data acquisition and curation, formal analysis, visualization, and original draft preparation. JG and his team supported field work and provided resources. FE and TB acquired funding and supervised the project. TM, AW, TB, JG, and FE all contributed in writing, reviewing and editing the manuscript.

*Competing interests.* The contact author has declared that none of the authors has any competing interests.

*Acknowledgements.* We would like to acknowledge the Team of the Chair of Aquatic Systems Biology for support during field work and provision of technical equipment, power access, and space. Our thanks also go to the Chair and Testing Office for Foundation Engineering, Soil Mechanics, Rock Mechanics and Tunneling, mainly Gerhard Bräu, for Loss On Ignition (LOI) measurements. Further, we are thankful to the Chair of Engineering Geology who made lab space and technical equipment for sediment analyses available. In addition, we would
like to thank Kai Zosseder und Daniel Bohnsack for guidance in thermal conductivity measurements, and Manuel Gossler for valuable input



on temperature measurements and modeling. We thank Theresa Mond and Sophia Klausner for their essential support during installation of the monitoring station, Jaroslava Obel for her help with laboratory analytics, and Friedhelm Pfeiffer for critical reading and reviewing of the manuscript.



## Appendix A: Sediment properties

For sediment characterization, cores were taken by manually pushing a liner with 6 cm inner diameter into the sediment. In September 2021 and August 2022 sieve-slurry analyses were performed, each time for two homogeneous layers, according to the German norm DIN 17892-4. Resulting grain-size distribution curves are displayed in Fig. A1. Porosity was measured from two separate liners by weighting a known volume of sediment before and after drying at 105 °C. The same samples were later used for the determination of organic carbon content as Loss On Ignition (LOI) according to the German DIN 18128.

After grinding and weighting, samples were annealed in a furnace at 550 °C to constant mass, cooled to room temperature in a desiccator, and weighted again. Results showed that the sediment at the sampling site consisted of 3 % gravel, 27 % sand and 70 % silt with a porosity of 81.5 % and an LOI of 21 %.

Three additional cores were used for measurements of thermal conductivity with the TCi-3-A Thermal Conductivity Analyzer and a Transient Line Source (TLS) (C-Therm, Fredericton, Canada). The sediment cores were taken in liners with 42 cm

diameter and sample heights between 25 and 30 cm. Measurements were conducted at a constant temperature of 8±1°C, close to true sediment temperatures, in a cooling room, and samples were pre-tempered for >12 hours. The line source with a sensor length of 15 cm was inserted vertically in the center of the sediment core and heated with 0.1 W. In most measurements, small deviations from the expected linear relation between the logarithm of time and the change in measured temperature were observed. Linear regression reached $R^2 = 0.972$ to $0.984$. Most likely, this was caused by inhomogeneities in the sample or

small rates of water drainage and consolidation during the measurement. Values for thermal conductivity $\lambda$ between 0.56 and 0.64 W m$^{-1}$ K$^{-1}$ were found. In this study, we used the median $\lambda = 0.60$ W m$^{-1}$ K$^{-1}$. This value lies well in the range of 0.20 to 0.70 W m$^{-1}$ K$^{-1}$ (mean: 0.51 W m$^{-1}$ K$^{-1}$) found by Dalla Santa et al. (2020) for unconsolidated material with an organic matter content of >5%.

## Appendix B: Oxygen sensor calibration

Calculation of dissolved O$_2$ concentrations from measured phase angles was based on the two-site quenching model of the Stern-Volmer equation (Eq. B1) (Carraway et al., 1991; Vieweg et al., 2013).

$$\frac{tan(\phi)}{tan(\phi_0)} = \frac{f}{1 + K_{SV}[O_2]} + \frac{1 - f}{1 + mK_{SV}[O_2]} \tag{B1}$$

with $\phi$ being the measured phase angle, $\phi_0$ the phase angle at 0% a.s., K$_{SV}$ the quenching constant as a function of saturation O$_2$ concentration, and f and m fit paramters. The parameters f, m, and K$_{SV}$(20 °C, lab air pressure) were estimated as best fit

for calibration measurements conducted at 7 different dissolved O$_2$ concentrations at 20 °C (Fig. B1 (a)).

Measured phase angles are temperature-dependent, thus compensation for field temperatures was necessary (Vieweg et al., 2013). For this, measurements were conducted at 0 % a.s. and 100 % a.s. at five and four environmentally relevant temperatures between 5 and 25 °C. The change of measured phase angle per Kelvin $\Delta\phi K_{\phi_0}^{-1}$ and $\Delta\phi K_{\phi_{100}}^{-1}$ at 0 % a.s. and 100 % a.s., respectively, was estimated with linear regression (Eq. B2, B3 and Fig. B1 (b)).




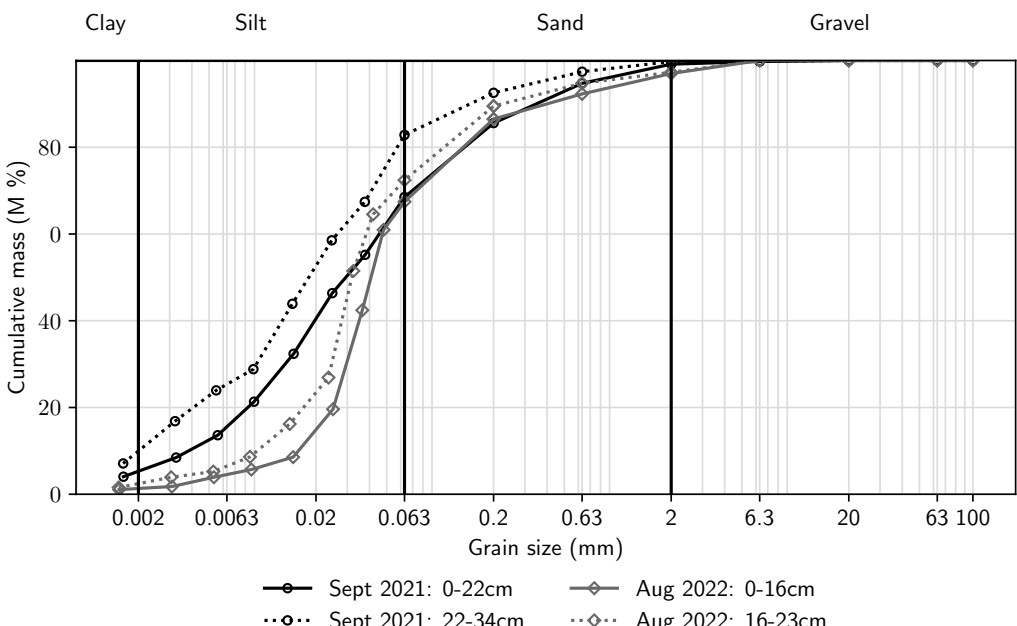

**Figure A1.** Grain-size distribution curves from sediment cores taken in September 2021 and August 2022.

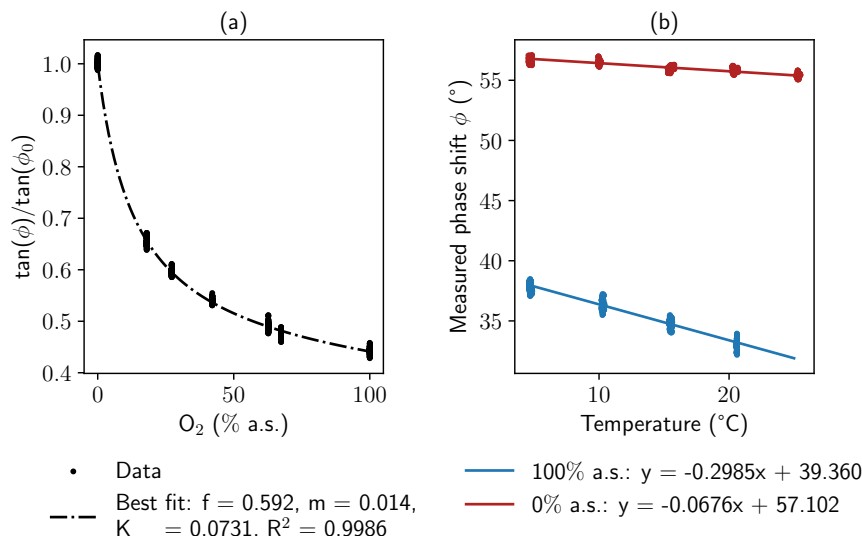

**Figure B1.** Calibration of the custom-made fiber-optical oxygen sensor. Panel (a) shows the Stern-Volmer Plot with best-fit parameters for the model and panel (b) the temperature dependence at 0 % and 100 % a.s.



$$tan(\phi_0)[T_m] = tan(\phi_0 + \Delta\phi K_{\phi_0}^{-1}(T_m - T_0)) \tag{B2}$$

$$tan(\phi_{100})[T_m] = tan(\phi_{100} + \Delta\phi K_{\phi_{100}}^{-1}(T_m - T_{100})) \tag{B3}$$

For the calculation of $O_2$ concentrations from phase angles measured in the field, first a fourth order polynomial was fit to temperature data recorded at the time of measurement to gain a continuous temperature depth-distribution (Fig. 3 (b)). Above

the sediment-water interface, average temperature of all sensors was assumed to be constant. For each depth, $K_{SV}$ was re-calculated based on $O_2$ saturation concentration, a function of water temperature and pressure at the specific depth. Then, $O_2$ concentrations were calculated with the Stern-Volmer equation (Eq. B1) in % a.s. and converted to $\mu mol\,L^{-1}$ based on depth-dependent saturation concentrations.

Due to the flat shape of the calibration model in saturated and near-saturated conditions (Fig. 3 (a)), small errors in measured

phase angles partly led to extremely high concentrations. To avoid these unrealistic values, all concentrations of >100 % a.s. were normalized such that the maximal concentration was 120 % a.s. (Eq. B4).

$$O_{2,nomalized} = \frac{20}{(O_{2,max} - 100)} \cdot (O_{2,original} - 100) + 100 \tag{B4}$$

where $O_{2,nomalized}$ is the normalized concentration value between 100 % and 120 % a.s., $O_{2,max}$ the maximally measured concentration considering all values of a profile, and $O_{2,original}$ the originally calculated concentration with an original value

of >100 % a.s.

**Appendix C: Additional pore-water analyses**

This section includes additional information on pore-water sampling and analyses. The equilibration period of the peeper was between April $6^{th}$ 2022 and May $3^{rd}$ 2022. Rhizon sampling at $0.19\,ml\,min^{-1}$ was conducted on May $3^{rd}$ right before sampling of the peeper. Pump rates of $0.09\,ml\,min^{-1}$ and $0.38\,ml\,min^{-1}$ were tested on May $30^{th}$ and $31^{st}$, respectively.

Figure C1 shows concentration profiles of $Ca^{2+}$, $Mg^{2+}$, and $Cl^-$ concentrations. The same data is displayed in box plots in Fig. C2 where also significant differences between sampling techniques are shown. Differences between peeper and Rhizon samples may be affected by the sampling technique or small-scale chemical heterogeneities since the peeper was placed approx. 15 cm away from the monitoring station to avoid mutual disturbances. Box plots are also provided for $CH_4$ concentrations and $\delta^{13}C$-$CH_4$ in Fig. C3, as well as $\delta^{18}O$ and $\delta^2H$ in Fig. C4. Data sets of $\delta^{18}O$ and $\delta^2H$ were not significantly different for high

and low pump rates.



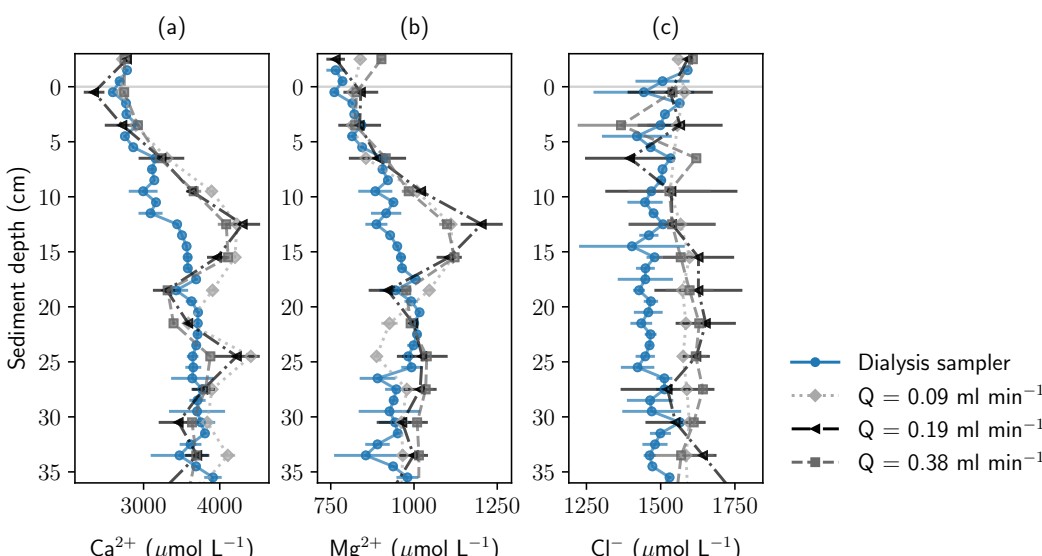

**Figure C1.** Concentration profiles of (a) $Ca^{2+}$, (b) $Mg^{2+}$, and (c) $Cl^-$ at four different sampling techniques. Error bars show standard deviation of repeated measurements (n=3).



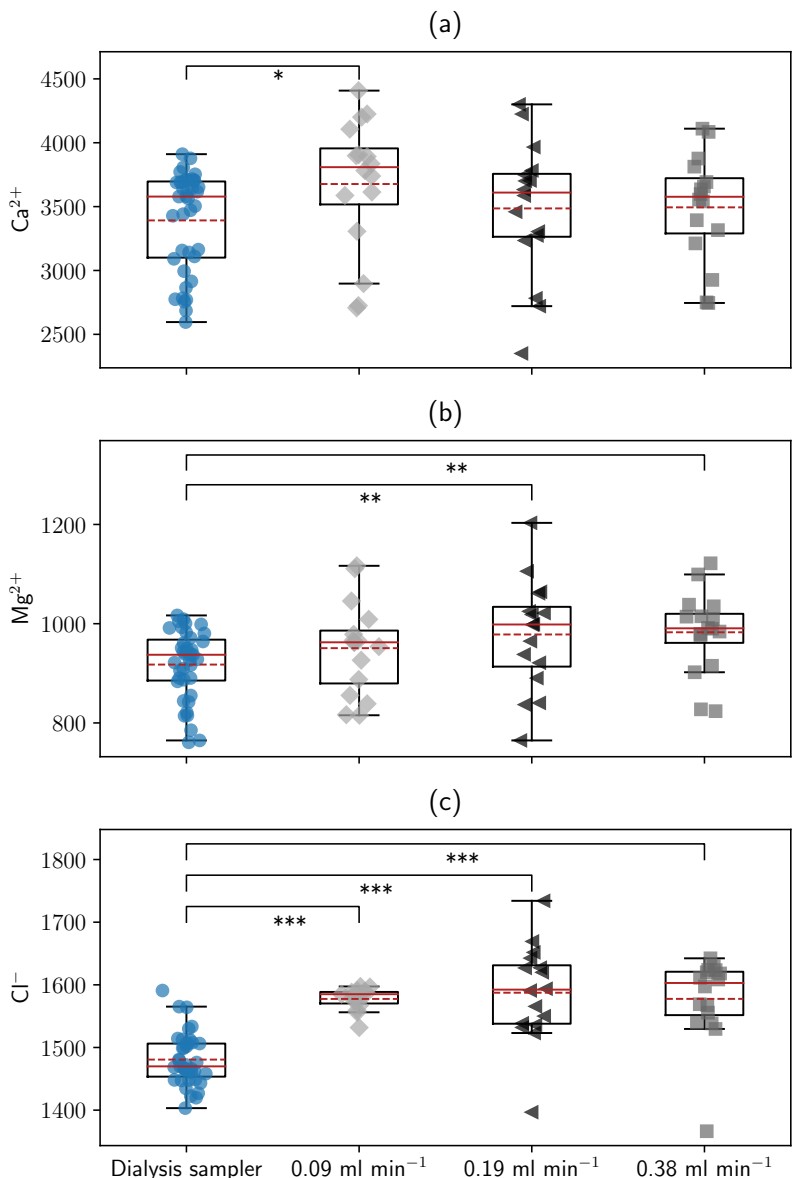

**Figure C2.** Box plots of (a) $Ca^{2+}$, (b) $Mg^{2+}$, and (c) $Cl^-$ concentration data. The box indicates the inter-qurtile range (IQR) between first and third quartile. Whiskers show 1.5 times the IQR. Median is displayed as solid, mean as dashed line. Where pairwise comparisons (Mann Whitney U test) showed significant differences, this is marked as follows: *(0.05 > p > 0.01), **(0.01 > p > 0.001), ***(p < 0.001).

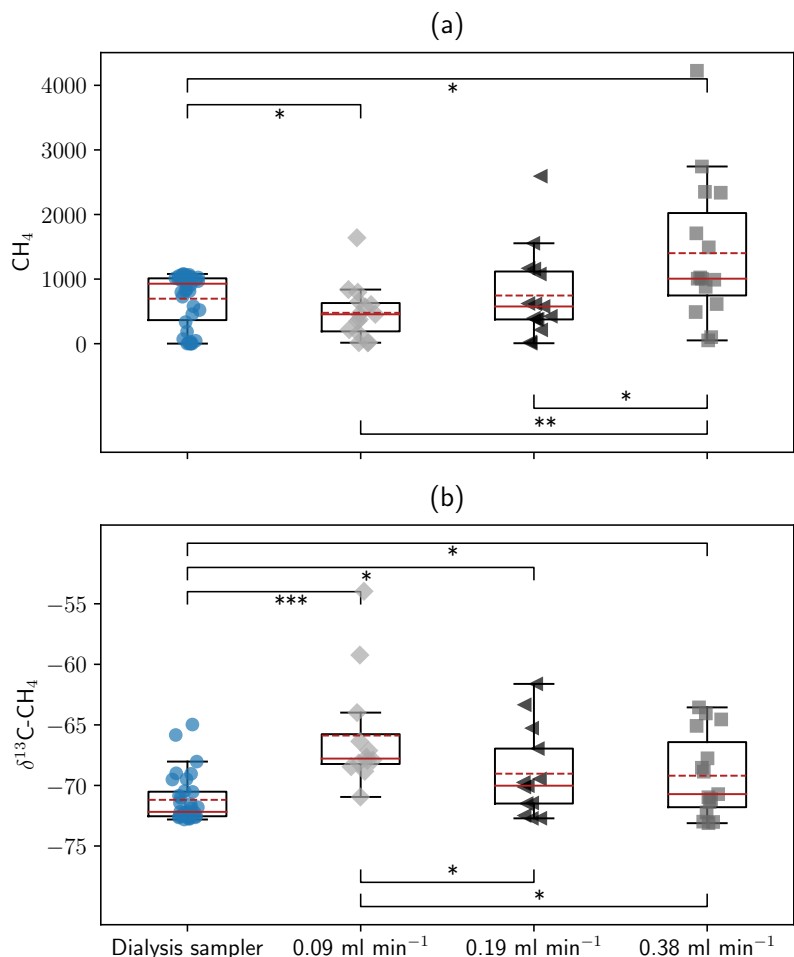

**Figure C3.** Box plots of (a) $CH_4$ concentration and (b) stable isotope measurements. The box indicates the inter-quartile range (IQR) between first and third quartile. Whiskers show 1.5 times the IQR. Median is displayed as solid, mean as dashed line. Where pairwise comparisons (Mann Whitney U test) showed significant differences, this is marked as follows: *(0.05 > p > 0.01), **(0.01 > p > 0.001), ***(p < 0.001).





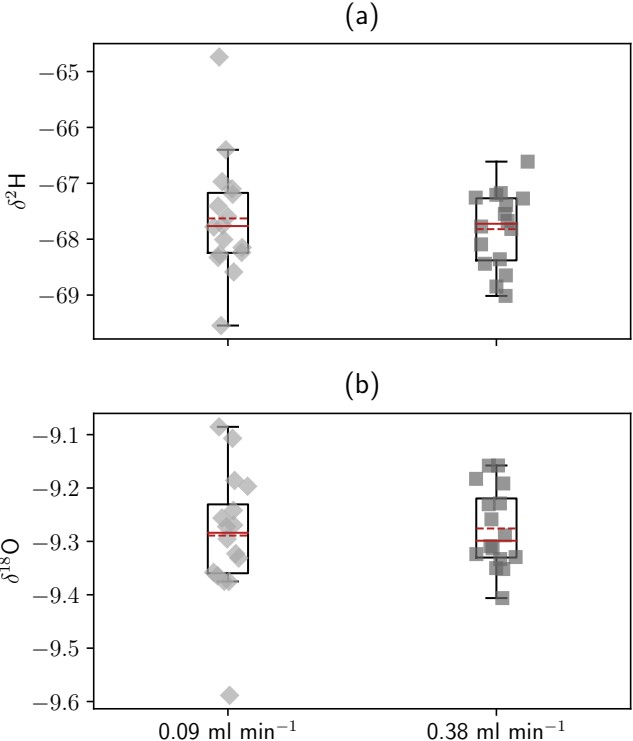

**Figure C4.** Box plots of (a) $\delta^2$H, and (b) $\delta^{18}$O data. The box indicates the inter-qurtile range (IQR) between first and third quartile. Whiskers show 1.5 times the IQR. Median is displayed as solid, mean as dashed line. Differences between the data sets were not significant.



## Appendix D: Detailed temperature modeling results

Flux rates calculated with both amplitude and phase methods by Hatch et al. (2006) and Keery et al. (2007) from the deepest 6 sensors in 6 cm, 8 cm, 10 cm, 12 cm, 14 cm, and 24 cm depth are given in Fig. D1. Fluxes were calculated between overlapping sensor pairs. For example, the flux calculated for 8 cm depth was calculated from the sensors in 6 cm and 10 cm depth. Mean,

mean of absolute values, range, and the percentage of negative values for each simulated time series are summarized in Tab. D1. Based on the amplitude method, the majority of values was negative when considering sensors at 8 cm depth and deeper, indicating upwards directed flow. Values calculated for shallower depths were mainly positive, showing large peaks when considering sensors placed in less than 6 cm depth. These peaks are assumed to be caused by sediment dynamics like sedimentation and erosion (see main paper). With the phase method, only absolute flux rates could be calculated. Fluxes calculated based on

phase change were 4-18 times larger than fluxes based on amplitude dampening.

| Depth | | Hatch amplitude | Keery amplitude | Hatch phase | Keery phase |
|---|---|---|---|---|---|
| 8 cm | mean | $6.3 \cdot 10^{-8}$ | $6.3 \cdot 10^{-8}$ | | |
| | mean (abs) | $1.7 \cdot 10^{-7}$ | $1.7 \cdot 10^{-7}$ | $3.0 \cdot 10^{-6}$ | $3.1 \cdot 10^{-6}$ |
| | range | $-5.6 \cdot 10^{-7}$ to $6.0 \cdot 10^{-7}$ | $-5.7 \cdot 10^{-7}$ to $6.0 \cdot 10^{-7}$ | $1.2 \cdot 10^{-6}$ to $5.7 \cdot 10^{-6}$ | $1.3 \cdot 10^{-6}$ to $5.7 \cdot 10^{-6}$ |
| | % < 0 | 34% | 34% | - | - |
| 10 cm | mean | $-1.6 \cdot 10^{-7}$ | $-1.6 \cdot 10^{-7}$ | | |
| | mean (abs) | $2.1 \cdot 10^{-7}$ | $2.1 \cdot 10^{-7}$ | $2.4 \cdot 10^{-6}$ | $2.5 \cdot 10^{-6}$ |
| | range | $-7.2 \cdot 10^{-7}$ to $4.5 \cdot 10^{-7}$ | $-7.3 \cdot 10^{-7}$ to $4.6 \cdot 10^{-7}$ | $4.4 \cdot 10^{-7}$ to $5.8 \cdot 10^{-6}$ | $1.5 \cdot 10^{-7}$ to $5.9 \cdot 10^{-6}$ |
| | % < 0 | 85% | 85% | - | - |
| 12 cm | mean | $-2.6 \cdot 10^{-7}$ | $-2.6 \cdot 10^{-7}$ | | |
| | mean (abs) | $2.8 \cdot 10^{-7}$ | $2.8 \cdot 10^{-7}$ | $1.8 \cdot 10^{-6}$ | $1.9 \cdot 10^{-6}$ |
| | range | $-7.9 \cdot 10^{-7}$ to $3.4 \cdot 10^{-7}$ | $-8.1 \cdot 10^{-7}$ to $3.5 \cdot 10^{-7}$ | $4.3 \cdot 10^{-7}$ to $4.4 \cdot 10^{-6}$ | $1.7 \cdot 10^{-7}$ to $4.4 \cdot 10^{-6}$ |
| | % < 0 | 90% | 90% | - | - |
| 18 cm | mean | $-4.9 \cdot 10^{-7}$ | $-5.0 \cdot 10^{-7}$ | | |
| | mean (abs) | $4.9 \cdot 10^{-7}$ | $5.0 \cdot 10^{-7}$ | $2.1 \cdot 10^{-6}$ | $2.1 \cdot 10^{-6}$ |
| | range | $-1.2 \cdot 10^{-6}$ to $-3.5 \cdot 10^{-8}$ | $-1.2 \cdot 10^{-6}$ to $-3.5 \cdot 10^{-8}$ | $4.3 \cdot 10^{-7}$ to $5.0 \cdot 10^{-6}$ | $2.4 \cdot 10^{-8}$ to $5.1 \cdot 10^{-6}$ |
| | % < 0 | 100% | 100% | - | - |

**Table D1.** Summary of results from VFLUX modeling from sensors in 6 cm, 8 cm, 10 cm, 12 cm, 14 cm, and 24 cm depths. Fluxes were calculated between each other sensor. For example, the flux calculated for 8 cm depth was calculated from the sensors in 6 cm and 10 cm depth. Lower sensors were not included due a strong influence of sedimentation and erosion events. All values are given in $\mathrm{m\,s^{-1}}$.



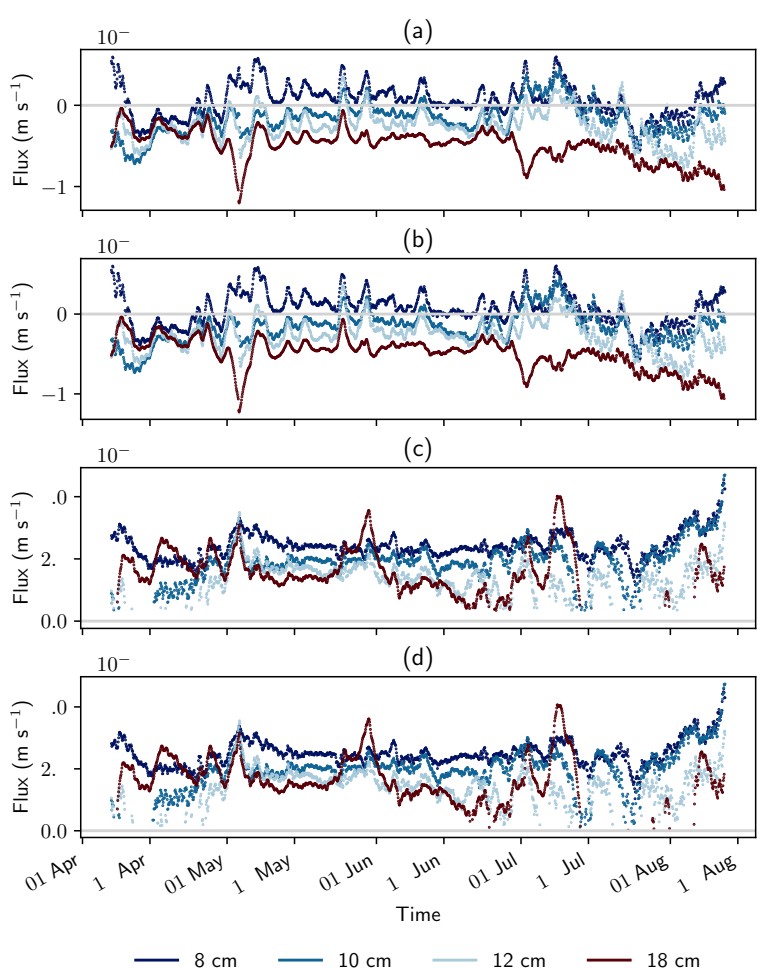

**Figure D1.** Detailed results of VFLUX modeling. Calculated fluxes are based on (a) amplitude method by Hatch et al. (2006), (b) amplitude method by Keery et al. (2007), (c) phase method by Hatch et al. (2006), and (d) phase method by Keery et al. (2007). Positive flow in (a) and (b) is downwards directed. The phase method in (c) and (d) only gives absolute values and no direction of flow.



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
