# Peer review of "Technical Note: Testing pore-water sampling, dissolved oxygen profiling and temperature monitoring for resolving dynamics in hyporheic zone geochemistry"

_EGUsphere, 2023_

## Author Comment (AC1)

Answer to Anonymous Referee #1

**Technical Note: Testing pore-water sampling, dissolved oxygen profiling and temperature monitoring for resolving dynamics in hyporheic zone geochemistry**

First of all, we would like to thank the referee for the positive review and the helpful comments. We are pleased that the manuscript was received as well written and worth publishing. We will answer the specific comments in detail below.

You use a thermal dispersivity of 0.001 m from the literature (table 1), which probably is a very rough estimation. Usually thermal dispersion is low in comparison to thermal conductivity, but this can be different in case with high water flux, such as yours. So, the question is whether thermal dispersion is relevant and, if so, it can affect the calculation of water fluxes. Can this issue be addressed? It may also be related to the next comment.

Thank you for addressing this point. It is true that thermal dispersivity is a rough estimate from the literature. To test the influence of the parameter on estimated hyporheic exchange fluxes, we performed a Monte Carlo analysis for the sensor pair in 8 cm and 12 cm depth, and for the first period of our data. 100 runs of VFLUX were performed for each scenario on the reduced data set. The thermal dispersivity parameter  $\beta$  was chosen as a random variable with normal distribution based on the following mean and standard deviation values:

| μ     | σ      |
|-------|--------|
| 0.001 | 0.0005 |
| 0.01  | 0.005  |
| 0.1   | 0.05   |

The data shows that a lower dispersivity leads to larger fluctuations and higher flux estimates compared to higher dispersivity values (Fig. D2). Calculated fluxes would be smaller in absolute terms and more stable if the diffusivity was larger than initially assumed. We will add Fig. D2 and the following text to App. D:

Figure D2: Monte Carlo analysis for thermal dispersivity. Flux values are given per unit area. Three scenarios were tested for mean and standard deviation of the thermal dispersivity parameter  $\beta$ . Results were generated with n=100 runs for each scenario. Shading indicates 95% confidence intervals for each scenario. The results were calculated with the software package VFLUX and the Hatch amplitude method.

"The influence of the thermal dispersivity parameter  $\beta$  was tested with a Monte Carlo analysis on a reduced data set, including data from April and May 2022 and the sensor pair in 8 cm and 12 cm depth. A normal distribution was assumed for the parameter  $\beta$ , with different means and standard deviations. For each scenario, 100 runs of VFLUX were performed with the random variations of  $\beta$  according to the respective distribution. The results show that higher thermal dispersion would lead to lower absolute flux values and less intense fluctuations (Fig. D2). Considering that  $\beta$  was changed by two orders of magnitude, the sensitivity of the model to changes in dispersivity appear to be limited. Nevertheless, further investigations on thermal dispersivity could help to improve the use of temperature measurements for hyporheic exchange flux modeling."

Appendix D presents the water fluxes calculated from the temperatures with various methods. Differences between results of the methods are quite high (4-18 times). How accurate are the results of figure 4? Can we compare these fluxes with some other measurement?

The differences you mentioned mainly occurred between the different methods (using amplitude dampening or phase change). Results were more consistent within one method. We suggest that using the amplitude method in our case is more reliable. Amplitude dampening is pronounced in the data while phase differences between the sensor pairs were very small. In fact, trying to compare neighboring sensors did not produce any result with the phase method. Unfortunately, there is no data to validate the findings.

We added the following paragraph to App. D to better explain the differences and reasons why we chose to present the amplitude data in the paper:

"Fluxes calculated based on phase change were 4-18 times larger than fluxes based on amplitude dampening. Amplitude dampening was pronounced in the data while phase differences between the sensor pairs were only very small. In fact, it was not possible to get flux estimates from neighboring sensors with the phase method due to the minimal time lag which was smaller than the temporal resolution of the time

series. Therefore, we hypothesize that for our data set estimates based on the amplitude method are much more reliable and have chosen not to display results based on the phase method in the main paper. The data is still displayed here to allow a comparison and for transparency by showing all results."

**A porosity of 81.5% (Table 1 and appendix A) is quite high. Is there some reason for this high value?**

Indeed, a porosity of 81.5% seems very high. We were surprised ourselves but found these high porosities in repeated measurements at different times. However, having porosities around and above 80% in fine, unconsolidated marine, organic-rich and lake sediments is not uncommon (Iversen & Jørgensen, 1993; Sweerts *et al.*, 1991). At our study site, the fine, organic-rich bed substrate was similar to marine or lake sediments. The site was located in a zone of reduced flow velocities due to a log lying crosswise a few meters upstream which allowed settling of very fine particles. The observed steep geochemical gradients (Fig. 2) confirm the similarity to marine or lake environments. The high porosity can also be explained by the lack of consolidation. Deposits were very loosely bedded and easy to stir as observed under light physical stress.

Why do you put profiles of Ca, Mg and Cl concentrations in appendix C and those of NO3 and SO4 in the body of the paper? I suggest, for coherence, to move the profiles of Ca, Mg and Cl to the body. The box plots can remain in the appendix.

Yes, for reasons of coherence, it makes sense to move the geochemical profiles of  $Ca^{2+}$ ,  $Mg^{2+}$  and  $Cl^-$  to the body of the paper. We will expand Fig. 2 to include these profiles.

---

## Author Comment (AC2)

Answer to Anonymous Referee #2

**Technical Note: Testing the effect of different pump rates on pore-water sampling for ions, stable isotopes and gas concentrations in the hyporheic zone**

The study tests and partly compares the analyses from four different techniques to evaluate hydro-biogeochemical processes in hyporheic zones. These techniques are a pore-water dialysis sampler (peeper), a pore-water Rhizon sampler (similar to MINIPOINTS and small multi-level piezometers), an in-situ dissolved oxygen profiler and a temperature profiler, all installed at one location. The techniques, which were directly compared to each other (the peeper with the in-situ dissolved oxygen profiler and the peeper with the Rhizon sampler), were compared to each other only once, respectively. In addition, the study quantifies the effect of three different pumping rates on the pore-water solute and gas concentrations withdrawn with the pore-water Rhizon sampler.

The study found that a) the peeper and the in-situ dissolved oxygen profiler gave comparable results for the dissolved oxygen concentration, b) the peeper and the Rhizon sampler gave comparable results for the ion concentrations and stable water isotopes but resulted in deviating $CH_4$ concentrations and the $\delta^{13}C$ signature of $CH_4$, and c) that the pumping rate to withdraw pore-water samples from the Rhizon sampler had an effect on the $CH_4$ concentration and its isotopic composition, but not on the other ions and the stable water isotopes.

We want to thank you for this very careful and detailed review. We really appreciate the amount of work and hope that you will find the revised manuscript more focused and concise. Some of your suggestions required additional paragraphs of text, figures, and tables. We tried to compensate the added content by removing some superfluous paragraphs from the main text. Some of the new content is added as an extension to the supplementary material.

Below, we have provided a detailed point-by-point list of answers and replies to the comments and suggestions raised by the reviewer. We have made every attempt to address the excellent suggestions and the numerous valuable recommendations where appropriate and have provided detailed responses and explanations below.

**Response to general statements**

The manuscript is very well written, with clear and complementary figures and tables. In addition, the technical, analytical and fieldwork efforts of the authors are very appreciable, knowing from my own experiences how delicate it is to study and sample the pore-water of the hyporheic zone.

Thank you for that general assessment.

However, the current focus of the manuscript provides only negligible advances of known experimental techniques. Nevertheless, I think, that some parts of the manuscript have potential and the updated manuscript should focus on them. The current focus of the manuscript is based on techniques that are not new (as stated by the authors themselves) and which have been used with the same or slightly different designs for several to many years to study the hydrobiogeochemistry of the hyporheic zone. Furthermore, the authors only directly compare two pairs of those techniques and do this only once each. It is for these reasons, that this manuscript in its current form is marginally novel and would only provide an incremental contribution. This is, even though the exact combination of those four techniques might be new and even though these techniques provide complementary information (which is neither new). If the authors wish to publish a research article, based on these techniques and a more extended dataset, they could describe these techniques directly in it, without the necessity of a prior Technical Note.

In contrast, the evaluation of the effect of the pumping rate (to extract pore-water from the Rhizon sampler) on the ion, isotope and gas concentration is novel and important. It has the potential to improve the current techniques and the interpretations based on hyporheic pore-water sampling. I would therefore suggest to strongly re-structure the manuscript (including the title) in order to put the focus on the effect of the pumping rate on the gas and the solute concentrations as well as the isotopic signatures. For that, I would suggest to remove the parts about the hyporheic oxygen and temperature sensors and focus on the Rhizon sampler with its varying sampling rate. The data of the peeper could be included as a comparison.

Thank you for highlighting the novelty of our overall experimental approach. These results certainly deserve more focus. This is now reflected in the title and the revised manuscript.

You are correct, that the methods for temperature monitoring and oxygen profiling are not entirely new and published elsewhere. We agree that a stronger focus on the novel aspects is useful, and we have adjusted the paper accordingly. However, we think that it is essential to also provide data on dissolved oxygen and temperature since this information is crucial for the site characterization as well as for the methodological comparison. These parameters are essential for hydrochemical data interpretation and the understanding of processes in hyporheic zone geochemistry. Furthermore, to the best of our knowledge the combination of these three methods has not been employed before. Oxygen measurements in samples withdrawn with Rhizon samplers or other techniques (USGS MINIPOINTS and/or multi-level piezometers) are error-prone: contamination with atmospheric oxygen is very likely and a precise distinction of oxic and anoxic zones is therefore not possible with these techniques. In-situ measurements are necessary, and the manufactured sensor is a very valuable tool for this, particularly where oxic and anoxic redox zones are very close together. Without the oxygen sensor, our installation with Rhizon samplers for pore-water extraction would be much less valuable for hyporheic zone characterization. We also agree that the use of temperature as a tracer for groundwater flow or hyporheic exchange has been discussed in the literature multiple times and is a well-studied subject. Yet, hyporheic exchange flux estimation is rarely combined with geochemical profiling. Making use of the sensors that were needed for evaluation of the $O_2$ sensor's raw data (see Supplement S2), we decided to monitor temperature as a proxy for changes between sampling campaigns.

In response to your suggestions, we reformulated the abstract to incorporate the new focus (see p. 1, lines 1-15 and the text below). We also highlighted synergies between the three methods in the discussion (see p. 16, line 359 ff. and the reply later in the discussion section of this document).

"The hyporheic zone (HZ) is of major importance for carbon and nutrient cycling as well as for the ecological health of stream ecosystems, but also a hot spot of greenhouse gas production. Biogeochemical observations in this ecotone are complicated by a very high spatial heterogeneity

and temporal dynamics. It is especially difficult to monitor changes in gas concentrations over time, because this requires pore-water extraction which may negatively affect the quality of gas analyses through gas losses or other sampling artefacts. In this field study, we wanted to test the effect of different pump rates on gas measurements and installed Rhizon samplers for repeated pore-water extraction in the HZ of a small stream. Pore-water sampling at different pump rates was combined with an optical sensor unit for in-situ measurements of dissolved oxygen, and a depth-resolved temperature monitoring system. While Rhizon samplers were found to be highly suitable for pore-water sampling of dissolved solutes, measured gas concentrations, here $CH_4$ showed a strong dependency of the pump rate during sample extraction, and an isotopic shift in gas samples became evident. This was presumably caused by a different behaviour of water and gas phase in the pore-space. The manufactured oxygen-sensor could locate the oxic-anoxic interface with very high precision. This is ecologically important and allows to distinguish aerobic and anaerobic processes. Temperature data could not only be used to estimate vertical hyporheic exchange, but also depicted sedimentation and erosion processes. Overall, the combined approach was found to be a promising and effective tool to acquire time-resolved data for the quantification of biogeochemical processes in the HZ with high spatial resolution."

In an updated version of the manuscript, with a focus on the effect of varying pumping rates on the analysed pore-water concentrations, the limitations of the experimental design and alternative interpretations of the results need be considered more thoroughly. The main limitation is that the Rhizon samples were withdrawn only once for each pumping rate with almost four weeks between the first and the second sampling date. The possibility, that the observed differences are therefore due to variable biological activities, for example, prior to the sampling moment and not due to the variable pumping rates, should be thoroughly discussed in the updated version.

The more detailed comments below are about the pore-water Rhizon and dialysis sampler, as I would suggest to remove the parts about the oxygen and temperture data.

We would like to answer to the concern that differences observed at different pump rates may be caused by changes in $CH_4$ concentration over time rather than the sampling technique and give some more information on the installation and original goal of the sampling station.

The sampling station was already installed in March 2021 with the aim to better understand hyporheic methane dynamics. $CH_4$ profiles observed during 11 sampling campaigns in summer 2021 were irregular with large concentration differences in consecutive depths and stable isotope values significantly different from what had been measured with dialysis samplers (peepers) earlier in the same river (Michaelis et al., 2022). In 2021, samples were withdrawn with syringe pumps and plastic syringes. We suspected that the poor quality of methane profiles was first, caused by long holding times in plastic syringes, potentially favoring gas losses, and second, due to the fact that not all samples could be withdrawn at the same time. We therefore revised the sampling method to use peristaltic pumps and gastight tubes directly connected to the gas vials to 1) ensure a gastight transfer of the sample into the vial and 2) allow simultaneous sampling of all 15 ports. We installed a peeper for a comparison to ensure good data quality. When we again discovered large differences between $CH_4$ profiles of peeper and Rhizon samplers, we decided to test the influence of different pump rates.

Despite a thorough literature research and contacts to other research groups we experienced these pitfalls. Due to the low quality of gas concentration and stable isotope measurements and the

need to describe a second sampling methodology, we initially did not include the 2021 data into the manuscript. However, since these aspects relate to practically important questions of how to take samples in the future and at other sites, we reevaluated the value of the data and now think that a presentation of these findings and their discussion could be useful.

The data can also provide an answer to the concerns of the referee with regard to the low number sampling campaigns. The measurements in 2021 show a very stable system with marginal and only slow changes in hyporheic zone geochemistry. Despite the irregular gas concentration and isotope profiles, average concentrations were similar between measurements. Methane concentration measurements from a peeper installed in autumn 2021 confirm that gas concentrations are very stable over time, since they show almost exactly the same profile as in spring 2022. Oxygen and temperature data further confirm the stability of the system. The oxic-anoxic interface was always found to be very steep and located directly at the sediment-water interface. Modeled hyporheic exchange fluxes showed stability in the deeper layers where $CH_4$ is typically produced. We are therefore very confident, that the observed differences at different pump rates are actually due to the sampling technique, and not due to condition changes at the site between the sampling campaigns.

**Answers to detailed comments**

**Abstract:**

L4 and L13: You refer to time-resolved measurements and/or high temporal resolution, but you are not showing that the Rhizon sampler is adequate for measurements in the hyporheic zone with a high temporal resolution (which is certainly also the case for other hyporheic sampling techniques). I would suggest to rephrase it.

Thank you, this was rephrased to (see p. 1, lines 13-15):

"Overall, the combined approach was found to be a promising and effective and powerful tool to acquire time-resolved data for the quantification of biogeochemical processes in the HZ with high spatial resolution."

**Introduction:**

L21 to L41: I think important techniques to withdraw pore-water are missing, when you are referring to previous techniques to sample hyporheic pore-water. I would suggest to add the description of and the references to scientific articles using the extensively used USGS MINIPOINTS (Duff et al., 1998; Knapp et al., 2017) and/or multi-level piezometers (Krause et al., 2012; Rivett et al., 2008; Schaper et al., 2018) (the references given are only examples), which are very similar to the Rhizon samplers you described (even though different in detail).

Thank you for this addition and especially the provision of many valuable references. We included the methods as follows (see p. 2, lines 48-56):

"Several methods have been developed and applied for direct pore water extraction from the HZ. For example, USGS MINIPOINTS consist of several steel drivepoints with different lengths for the extraction of pore-water from several depths (Duff et al., 2017). In a similar way, depthresolved hyporheic pore-water sampling has been realized with multi-level piezometers, a set of tubes with different types of screens at the tips (Krause et al., 2012; Rivett et al., 2008; Schaper et al., 2018) or with fixed PVC tubes attached to syringes (Geist & Auerswald, 2007). Rhizon samplers (microfilter tubes), typically applied for soil moisture measurements in the unsaturated zone, have also occasionally been used for pore water extraction: Shotbolt (2010) used Rhizon samplers for pore-water extraction from sediment cores, Seeberg-Elverfeldt et al. (2005) in combination with an in-situ chamber in the Wadden sea, and Song et al. (2003) to sample pore-water from lake sediment microcosms. From each of these systems, samples can either be withdrawn with syringes or peristaltic pumps (Knapp et al., 2017; Seeberg-Elverfeldt et al., 2005)."

In the updated version of the manuscript, with a focus on the effect of the sampling/pumping rate on the measured solute and gas concentrations, it could be useful to consider/discuss the study of (Duff et al., 1998), as it has been referenced frequently to justify the pumping rates to withdraw hyporheic pore-water (a complete version of the paper is accessible at the USGS: https://water.usgs.gov/nrp/jharvey/pdf/l&o_1998_v43(6)_p1378.pdf).

To address the comment, we have now changed the last section of the introduction including aims and scope. As suggested by the referee, the focus is now more on the test of Rhizon samplers for gas analyses in pore water and especially the test of different pump rates. The above-mentioned source was included in the revised section which now reads: (see p. 2, line 57 ff.)

"However, these methods have rarely been used for gas analyses in hyporheic pore-water. Negative pressure can lead to outgassing and therefore, when pulling out the samples, gas contents may get affected. Suitable pump rates for pore-water extraction have been evaluated from chloride gradients, and rates $< 4.0$ ml min$^{-1}$ were found to be acceptable (Duff et al., 1998). But the effect of pump rates on gas concentrations has never been tested. Especially in fine-grained bed substrates, where the pressure in the extraction system to maintain these flow rates has to be much lower than ambient pressure, degassing effects are no longer negligible. Gas concentrations will reflect the low pressure in the extraction system, which is very hard to measure. In this study, we wanted to test this hypothesis and installed a monitoring station at a site with fine-grained deposits close to the river bank where high methane ($CH_4$) concentrations were to be expected. 15 Rhizon samplers were installed with 3 cm vertical distance for repeated pore-water sampling. Three different pump rates for pore-water sampling were tested and the results were compared to geochemical profiles observed with a peeper that was installed very close to the Rhizon samplers.

The sampling station was amended with a custom-coated fiber-optical oxygen sensor unit based on the description of Brandt et al. (2017) for a precise allocation of the oxic-anoxic interface. Air contamination during sample extraction from sediment cores, peeper chambers, or other types of in-situ samplers is likely and problematic for studying anoxic processes. An in-situ sensor was therefore essential for the assessment of methane in the HZ. As a third component, temperature monitoring in 14 different depths was used for an estimation of hyporheic exchange. Flux rates were calculated with analytical models introduced by Hatch et al. (2006) and Keery et al. (2007) using the software package VFLUX (Gordon et al., 2012). The temperature data was also needed for evaluating raw data of the oxygen sensor."

**Methods:**

In general, I think adding more, but concise, information about when and how the samplers were installed, when (date and approximate time of the day) and how often (once) they were sampled and/or how long the period was between installation and first sampling, would be very useful. Some of these informations can be eventually found in various parts of the manuscript (tables, figure captions, appendix), but I would suggest to provide these details together in the method section. If the authors prefer to put/keep this information in the appendix, a reference to it should be added in the main text.

We have now included more details on the installation and first sampling campaigns in 2021. We specified sampling dates etc. in the methods section and added a supplementary chapter (Section S2) on the 2021 data. The additional data was mentioned in the methods section as follows (see p. 3, lines 85-90):

"The monitoring station was installed on March 15th, 2021. For installation, a protective casing was manually pushed into the stream bed, the interior of the casing was cleared of sediment to allow the sampler to be inserted without damaging the filter tubes or temperature sensors, and finally the protective casing was removed and the sampler left to settle in. After installation, we observed heavy sedimentation and during the summer months, mainly between July and September, major macrophyte growth. The first sampling campaign was done two weeks after installation, when disturbances caused by the installation had been wearing off. 10 more sampling campaigns were performed in 2021, three in 2022 (Sec. S1, Tab. S1)."

Results were shortly described in the revised manuscript as well (see p. 10, lines 258-264):

"In addition, the hyporheic geochemistry of the study site was described in detail with 11 sampling campaigns between April and September 2021 (App. B). Geochemical gradients were found to be very steep, with oxygen reduction and denitrification zones in close proximity or even partly overlapping. A substantial amount of $CH_4$ was produced in the deep anoxic layers of the HZ. Ion and gas concentrations were stable over time with only gradual changes between spring and summer. The most pronounced changes were sedimentation events which moved the location of the sediment-water interface upwards. The anoxic, reduced conditions in deeper layers stayed unchanged throughout the sampling period in 2021. $CH_4$ concentration profiles measured with a peeper in September 2021 and in May 2022 showed almost exactly the same gradients."

**Appendix B: Geochemistry of the study site**

During 11 sampling campaigns between April and September 2021, samples were withdrawn with two LA-110 High Pressure syringe pumps (HLL Landgraf Laborsysteme, Langenhausen, Deutschland) at a pump rate of 0.15 mL min$^{-1}$. Dates, sampling method and pump rate for all

sampling campaigns are summarized in Tab. B1. The syringe pumps were equipped with 3D printed racks to hold 5 syringes each. Thus, up to 10 samples could be withdrawn simultaneously. Samples were collected in the syringes and then transferred to the respective vials for gas, sulfide, anion, or cation analyses. However, several disadvantages became obvious during sampling: not all 15 Rhizon samplers could be sampled simultaneously, thus making cross-contamination of samples from different depths more likely; syringes filled at different speeds, potentially due to sediment heterogeneities and gas intrusions; long stay of the sample in the syringes during collection made gas losses more likely. Therefore, the sampling technique was improved in 2022 as described in the main text.

Sample collection was carried out as described in Sec. 2.1. For gas sampling with syringe pumps, two needles were pierced through the rubber stoppers for sample injection, one connected to the syringe and one for pressure exchange. Samples were injected slowly along the side of the vial to prohibit degassing. Both needles were removed directly after sampling.

| Date | Sampling technique | Pump rate |
|---|---|---|
| 19.04.2021 | Rhizon samplers + syringe pumps with space for max. 10 plastic syringes | 0.15 mL min$^{-1}$ |
| 10.05.2021 | | |
| 26.05.2021 | | |
| 09.06.2021 | | |
| 23.06.2021 | | |
| 06.07.2021 | | |
| 20.07.2021 | | |
| 03.08.2021 | | |
| 17.08.2021 | | |
| 01.09.2021 | | |
| 23.09.2021 | | |
| 23.09.2021 | Peeper | - |
| 03.05.2022 | | |
| 03.05.2022 | Rhizon samplers + peristaltic pumps and gastight tubing | 0.19 mL min$^{-1}$ |
| 30.05.2022 | | 0.09 mL min$^{-1}$ |
| 31.05.2022 | | 0.38 mL min$^{-1}$ |

Table S1: **Summary of sampling dates, measurent technique and pump rate.**

[Figure]

Figure S2: **Comparison of two depth-profiles measured with pore-water dialysis samplers (peepers) in September 2021 and May 2022.**

[Figure]

Figure S3: **Concentration- and stable isotope measurements conducted at the monitoring station during spring and summer 2021**. Panels on the left show concentrations over time as contour plots. Panels on the right show two selected depth-profiles.

L66: What is meant with *stable hydrological conditions*? Provide details. Furthermore, provide some details about the catchment size and/or the stream discharge during the experimental period.

This was specified in the following way (see p. 3, lines 78-81):

"The river Moosach is characterized by very uniform flow conditions due to regulations of the water level by weirs. This lack of dynamics is also considered one of the reasons for its stable stream bed material with high rates of fine sediment deposition (Auerswald & Geist, 2008). The area where the sampling site was situated upstream of a weir that keeps the headwater level nearly constant at all discharge conditions."

L68 and L357/L362 (Appendix A): Which value for the sediment density was used to calculate the porosity? Was the relatively high organic matter content considered in the calculation? If not you might want to have a look here (Adams, 1973; Rühlmann et al., 2006).

Porosity was calculated as quotient of the volume of the pore space over the total volume:

$$n = \frac{V_{porespace}}{V_{total}}$$

The pore space volume was calculated as the weight difference between and after drying at 105°C divided by the density of water:

$$V_{porespace} = \frac{G_{wet} - G_{dry}}{\rho_{water}}$$

The porosity calculations were therefore independent of the sediment density.

Sediment density was only used to obtain the grain-size distribution curves shown in A1. We used a sediment density of 2.20 g/cm$^3$. This lies between 2.24 g/cm$^3$ based on Adams (1973) and 1.94 g/cm$^3$ based on Rühlmann et al. (2006), using a content of organic matter of 21% (see App. A).

L70: *After installation*: when? For how long? How? Did you encounter any difficulties during installation? Provide details.

We added details of the installation process to the revised manuscript (see p. 3, lines 85-90):

"The monitoring station was installed on March 15[th], 2021. For installation, a protective casing was manually pushed into the stream bed, the interior of the casing was cleared of sediment to allow the sampler to be inserted without damaging the filter tubes or temperature sensors, and finally the protective casing was removed and the sampler left to settle in. After installation, we observed heavy sedimentation and during the summer months, mainly between July and September, major macrophyte growth. The first sampling campaign was done two weeks after installation, when disturbances caused by the installation had been wearing off. 10 more sampling campaigns were performed in 2021."

L82: Did you ever encountered problems due to clogging, when using the Rhizon sampler with a pore-size of 0.1-0.2μm?

We did encounter problems right at the beginning when a biofilm grew on the upper three filters that lay above the sediment-water interface and blocked them completely. After changing these three filters on June 7th, 2021, this problem did not re-occur, presumably due to shading by leaves and water plants that re-grew after installation. No problems with clogging occurred at samplers within the sediment. To avoid potential clogging, 2 ml pore-water were flushed back after each sampling campaign. Care was taken to only flush back water that was prior removed from the pore-space and left in the sampling tubes. All samplers remained functional over a long time period. However, actual sampling speed differed for the different samplers despite equal sampling conditions. We attributed this observation to the presence of gas bubbles. Gas was observed in the tubes during pumping.

This was included to the manuscript as follows (see p. 4, lines 101-104):

"Clogging of the Rhizon samplers with a pore size of 0.1-0.2 μm occurred only once shortly after initial installation at three samplers above the sediment-water interface due to biofilm growth. After replacing the top three samplers, this problem did not re-occur. No problems with clogging occurred at the samplers within the sediment. To avoid potential clogging, 2 ml of pore water still in the sampling tubes after each sampling campaign was backwashed."

L86 – L89: Provide details about when (date + approximate time if it was different for the three sampling dates) the three pore-water extractions were conducted and which pump rate corresponds to which sampling date. State the three pump rates here (and not only the two extremes) and how long the pumping-rate dependent pore-water extractions lasted (almost 2 hours for the lowest pump rate?). Correct the lowest pump rate (L88: 0.01 ml/min). Did you rinse the tubes before each withdrawl to avoid that you are collecting water which has been stagnant in the sampling tube? If yes, how many ml?

We added a summary in Tab. S1 and the following information to the manuscript (see p. 5, lines 112-114):

"Three pumping rates were tested: 0.09 ml min$^{-1}$ on May 30th, 0.19 ml min$^{-1}$ on May 3rd, and 0.38 ml min$^{-1}$ on May 31st. Prior to sampling, 4 ml of pore-water were taken for pre-rinsing to exchange at least the tube volume of 3.8 ml without increasing the extracted volume too much."

L101-L103: When was the peeper installed/sampled? How was it installed?

This information was added to the manuscript as follows (see p. 6, line 131 f.):

"Over a period of one month, between April 3rd and May 3rd, an equilibrium between the water in the chambers and the surrounding pore-water was obtained."

The length of the sampling period is also displayed in the new Fig. 2, see answer below.

L122: Were K and Na not analysed with the ion chromatograph? If yes, but not reported here, why?

Sure, they were also analyzed. We decided only to show measurements that can improve the data interpretation and avoid to not overload the paper with figures and data which are not discusses in the text. Therefore, we suggest not to add in the data in the revised manuscript. Nevertheless, we plotted the data below for reasons of transparency.

[Figure]

**Results:**

The axes texts and/or titles of several figures are not displayed correctly or units are missing. For example: Fig. 5 (x- and y-axis text), Fig. A1 (y-axis text), Fig. C2 – Fig. C4 (units on y-axis title are missing).

Thank you for pointing this out. We are sorry for the display problems in Fig. 5 and Fig. A1, the original figures were being displayed correctly. We will add units to the y-axis in Fig. C2-C4.

L221/L222: Why were both statistical tests performed? If the data fullfilled the requirements for the parametric t-test, why did the authors conduct the non-parametric test as well?

It is correct that the parametric t-test in this case would be sufficient. We changed this to (see p. 10, line 248 f.):

"Results were found to be similar with no significant differences based on the t-test."

L226: The stated increased variance in the stable isotope measurements of $CH_4$ is not obvious from Fig. 2 and Fig. C3. If the authors have conducted a statistical test, I would suggest that they provide details about it.

We have rewritten the paragraph (see p. 10, lines 252-259):

"With an average of -71.2 ‰ $CH_4$ had a significantly lighter isotopic composition in peeper samples compared to samples extracted with Rhizon samplers (averages between -65.9 ‰ and -69.2 ‰). The stable carbon isotopic composition of $CH_4$ was with -65.9 ‰ most heavy at the lowest pump rate. Homogeneity of variances was neither given in $CH_4$ concentration nor stable isotope data. Standard deviation of $CH_4$ concentrations increased with increasing pump rate (420 µmol $L^{-1}$ at the lowest, 678 µmol $L^{-1}$ at the mid, and 1119 µmol $L^{-1}$ at the highest pump rate), but was more similar for isotopic data. When comparing all four data sets with the Kruskall Wallis H test, differences were significant for both $CH_4$ concentrations ($p = 0.01$) and stable isotopes ($p = 0.0003$)."

Figure 5: Is the caption missing? What is visualized with the shadow around the red, horizontal line? Provide details.

The figure caption was extended to:

"Relation of $CH_4$ concentrations and isotopic composition. The average ± standard deviation of $\delta^{13}C$-$CH_4$ for all data points with concentrations > 950 µmol $L^{-1}$ (-72.0 ± 1.1 ‰) is shown in red."

Finally, I think it could be very useful to add at least a hydrograph and a timeseries of the water or sediment temperature for the experimental period. This would facilitate the comparison (and the interpretation) of the three sampling days.

Thank you for this good suggestion. We added the Figure below as Fig. 2 to the methods section. It shows stream temperature and discharge at a monitoring station approximately 5 km downstream of the sampling site. Discharge data was retrieved from the Bavarian State Office of the Environment (2023).

[Figure]

Figure 2: **Discharge and stream temperatures during the sampling period.** Discharge data from a monitoring station approximately 5 km downstream was retrieved from the Bavarian State Office of the Environment. The span between minimum and maximum discharge is shaded in light blue, average stream discharge is shown as a blue line. The equilibration period of the peeper is highlighted with grey background color. Vertical lines show sampling dates at the monitoring station and are coded to the sampling rates.

**Discussion:**

I think, the main limitation of that part of the manuscript, which is investigating the potential effect of the sampling/pumping rate on the solute and gas concentrations is, that each pumping rate has only been conducted once. It is, therefore, difficult to interpret, whether the observed differences on the three sampling days are due to the different pump rates (i.e., an experimental artefact) or due to real differences in the pore space. Real differences (in contrast to experimental artefacts) in the gas concentration of the pore-water could be, for example, due to contrasting water/sediment temperatures (Comer-Warner et al., 2018; Duc et al., 2010; Emerson et al., 2021) and/or potentially the hydrological conditions or a particular ebullition event releasing $CH_4$ suddenly from the sediment (?) during and prior to the sampling days. This limitation needs be clearly addressed and thoroughly discussed in the updated version of the manuscript.

You are right that this did not become fully clear from the information provided in the manuscript and we therefore decided to provide additional information on the sampling site in a new App. B. As mentioned above, the data from 2021, as well as temperature and dissolved $O_2$ gradients showed that hyporheic zone geochemistry at the sampling site was stable over time and subject only to rather small and slow changes. We integrated this argumentation into the manuscript as follows (see p. 13, lines 299-305):

"Based on the data from 2021, that showed a very stable geochemical system, rapid changes in stream geochemistry between the sampling days at the beginning and end of May 2022 are not expected. The stream temperature was very similar on all sampling days, and river discharge was only 0.09 $m^3$ $s^{-1}$ (4.8 %) higher at the end of the month (Fig. 2). Ebullition occurred sporadically, but no larger, sudden gas releases were observed at the sampling site, neither in 2021 nor during recent field campaigns. Therefore, a rapid change of gas concentrations in the sediment seems to be very unlikely and the observed changes in $CH_4$ concentrations and stable isotopic composition in $CH_4$ are most likely caused by the changes in pump rate and not by varying hydrological or geochemical conditions at the sampling site."

From the provided temperature time series, it is impossible to asses the stream and/or pore-water temperatures prior and during the sampling days. In addition, the authors do not provide information about the discharge conditions during the experimental period. It is therefore not possible to evaluate, whether the hydro-climatological conditions were similar for the three sampling dates. As mentioned above, I would therefore suggest to provide this information.

Added as Fig. 2.

A good point is that the experiments were not conducted in the order low – intermediate – high pump rate, but that they were mixed, which is in contrast to the results that show a gradient from low – intermediate – high pump rate.

Thank you.

L250/251: Except for Cl, which showed consistently higher concentrations in the Rhizon sampler, compared to the peeper (Fig C2), and for Mg under intermediate and high pump rates.

The exception was added to the sentence (see p. 13, lines 289-292):

"Our results showed an excellent agreement for ion concentration and stable water isotope measurements in pore-water samples for the two different methods used, and equally good agreement for different pump rates when using Rhizon samplers and peristaltic pumps. The only exceptions were $Cl^-$ concentrations, which were consistently higher at the monitoring station compared to the peeper, and $Mg^{2+}$ at medium and high pump rates (Fig. C2)."

L257 - 262: Do the authors have any indication for this hypothesis? Have air bubbles been observed? Has this been reported before? Can the authors provide evidence/references for their statement that $CH_4$ bubbles likely exist in the porespace? In addition, additional potential explanations should be discussed (e.g., observed differences are not sampling artefacts but real variations; different pump rates sample pore-water from different pore spaces?)

As mentioned above, air bubbles were observed sporadically. A study on ebullition approximately 650 m downstream showed significant $CH_4$ ebullition. Entrapped gas could also be seen in sediment cores and gas rose up through the tubes during sampling. The study on ebullition is unfortunately not published yet and we can therefore not cite it. We will, however, mention the observation of rising gas bubbles from the stream bed and the occurrence of gas in sediment cores.

We added and rewrote the following section to comprehensively discuss the different possible explanations for the observed behavior (see p. 13-14, lines 306-321):

"Of course, actual changes of gas content and composition between sampling days would explain the measured differences. If not triggered by temperature changes or discharge peaks, these could be caused by physical stress or a sudden ebullition event. However, these events seem rather unlikely considering the stagnating geochemistry in 2021 and the rather remote location of the sampling site without public access. More convincing seems the possibility that water is sampled from different parts of the pore-space at different pump rates. Pressure gradients around the samplers will change if the pump rate is increased.

Another possible explanation for the observed differences in $CH_4$ concentrations and carbon stable isotopic composition may be differing behaviors of water and gas phases in the interstitial pore space. Rising air bubbles were sporadically observed at the sampling site and entrapped gas was found in sediment cores. During sample extraction, gas was seen to travel upwards through the tubes. These gas bubbles might get trapped in front of the microfilters at low pump rates, because low negative pressures may not be sufficient for extraction of gas bubbles from the sediment. At higher pump rates, bubbles seem to get mobilized from a larger distance, potentially further away than liquid pore-water samples. Additionally, higher pump rates lead to greater negative pressures which may cause increased out-gassing and thus, creation of additional gas bubbles. Since the tubes were directly connected to the sampling vials, bubbles were not lost, but gas and water phase were both contained in the sample vial. This could explain the large scatter and high concentration peaks observed at higher pump rates. Most likely a combination of this effect and the extraction of sample from different parts of the pore-space is responsible for the observed differences in gas samples at different pump rates."

L289/290: *Advantages are the possibility for time-resolved measurements*: That does not differentiate the Rhizon sampler from well-established methods (MINIPOINT; multi-level piezometer). I suggest to rephrase.

We have now restructured the discussion and removed the paragraph in line 288-293. Instead, we want to discuss a comparison of Rhizon samplers to other methods such as MINIPOINTS and multi-level piezometers (see p. 15, lines 334-340):

"Other techniques for pore-water extraction such as multi-level piezometers or USGS MINIPOINTS were not tested in this study but may have similar advantages and disadvantages to Rhizon-samplers. They allow time-resolved measurements and are hypothesized to be better suited for measuring effect and distribution of gas in sediments than dialysis samplers. But if, as suspected, changes in negative pressure at different pump rates lead to a different behavior of gas- and water phase in the pore-space, this effect is likely to occur whenever samples are directly extracted from the pore-space, no matter with which device. Larger pore-diameters could increase the suitability for gas sampling, but we would still recommend testing the effect of different pump rates when working with gas analyses in this type of fine-grained environments."

L290/291: *..effects and distribution of gas bubbles in the pore-space become visible*: – This is one potential interpretation and the authors have not provided evidence to support it. I suggest to rephrase it and/or to add references.

As mentioned above, this paragraph was removed during revision. It was meant to summarize what was discussed above, but in the new version seems to be out-dated.

L313/L314: In theory, the Rhizon sampler could be used during/after floods. In practice, however, conducting porewater sampling during or shortly after floods is challenging and the authors have not provided evidence that their installation remains in place undisturbed during/after floods.

There were no larger floods during the time of installation, and we therefore cannot proof if the station would endure such conditions. Further testing of flood stability would be out of scope for this work. We added a remark that additional fastenings may be necessary in such a scenario (see p. 17 lines 374-376):

"The combination could, for example, be very useful for studying the effect of floods and droughts on stream ecosystems in terms of nutrient cycling and GHG emission pulses, although additional fastenings may be necessary to ensure stability during floods."

L320/L321: Again, this does not differentiate the Rhizon sampler from the existing methods (MINIPOINTS, multi-level piezometers) and is less novel than what the authors suggest.

The last section (line 317-325) will be removed from the discussion.

L325: Combining depth-resolved temperature measurements with measurements of the pore-water geochemistry is less novel than suggested by the authors, for example (Briggs et al., 2013).

The last section (line 317-325) will be removed from the discussion.

At the end of the discussion, highlighted the benefit of using the described methods for oxygen profiling and temperature measurements, because this apparently did not become clear in the submitted version of the manuscript (see p. 16, line 359 ff.):

"Dissolved $O_2$ concentrations measured in peeper chambers were elevated compared to in-situ measurements and we did not find an affordable way to measure dissolved $O_2$ concentrations in extracted pore-water samples without contamination with atmospheric air. Considering the steep geochemical gradients, the employed sampling resolution of 3 cm would not have been sufficient to precisely locate the oxic-anoxic interface. For the assessment of $CH_4$ in a case like this, there is a necessity for in-situ measurements. The sensor developed by Brandt et al. (2017) was a low-cost effective tool and a great addition to the monitoring station. Temperature sensors that were necessary for the evaluation of the $O_2$ sensor's raw data could also be used for a continuous monitoring of the sampling site. The data was used to describe the site as an upwelling system, which is important information for the interpretation of geochemical profiles, and in addition, could visualize sedimentation and erosion processes. The measurements could further help to improve geochemical transport models if applied, because diffusion coefficients are temperature dependent. However, the installation of the sensors must be done carefully to ensure a long service life. At our field site, several sensors stopped functioning properly, most likely due to problems at soldered joints and connectors, or due to humidity and water intrusion."

**References:**

Adams, W. A. (1973). THE EFFECT OF ORGANIC MATTER ON THE BULK AND TRUE DENSITIES OF SOME UNCULTIVATED PODZOLIC SOILS. *Journal of Soil Science*, *24*(1), 10-17. https://doi.org/10.1111/j.1365-2389.1973.tb00737.x

Auerswald, K. and Geist, J. (2018): Extent and causes of siltation in a headwater stream bed: catchment soil erosion is less important than internal stream processes, Land Degradation & Development, 29, 737–748, https://doi.org/10.1002/ldr.2779.

Bavarian State Office of the Environment (2023): Gewässerkundlicher Dienst Bayern, Data and information, https://www.gkd.bayern.de/en/, last access: 13 June 2023.

Brandt, T., Vieweg, M., Laube, G., Schima, R., Goblirsch, T., Fleckenstein, J. H., & Schmidt, C. (2017). Automated in situ oxygen profiling at aquatic–terrestrial interfaces. Environmental Science & Technology, 51(17), 9970-9978.

Briggs, M. A., Lautz, L. K., Hare, D. K., & González-Pinzón, R. (2013). Relating hyporheic fluxes, residence times, and redox-sensitive biogeochemical processes upstream of beaver dams. *Freshwater Science*, *32*(2), 622-641. https://doi.org/10.1899/12-110.1

Comer-Warner, S. A., Romeijn, P., Gooddy, D. C., Ullah, S., Kettridge, N., Marchant, B., Hannah, D. M., & Krause, S. (2018). Thermal sensitivity of CO2 and CH4 emissions varies with streambed sediment properties. *Nature Communications*, *9*(1), 2803. https://doi.org/10.1038/s41467-018-04756-x

Duc, N. T., Crill, P., & Bastviken, D. (2010). Implications of temperature and sediment characteristics on methane formation and oxidation in lake sediments. *Biogeochemistry*, *100*(1-3), 185-196. https://doi.org/10.1007/s10533-010-9415-8

Duff, J. H., Murphy, F., Fuller, C. C., Triska, F. J., Harvey, J. W., & Jackman, A. P. (1998). A mini drivepoint sampler for measuring pore water solute concentrations in the hyporheic zone of sand-bottom streams. *Limnology and Oceanography*, *43*(6), 1378-1383. https://doi.org/10.4319/lo.1998.43.6.1378

Emerson, J. B., Varner, R. K., Wik, M., Parks, D. H., Neumann, R. B., Johnson, J. E., Singleton, C. M., Woodcroft, B. J., Tollerson, R., Owusu-Dommey, A., Binder, M., Freitas, N. L., Crill, P. M., Saleska, S. R., Tyson, G. W., & Rich, V. I. (2021). Diverse sediment microbiota shape methane emission temperature sensitivity in Arctic lakes. *Nature Communications*, *12*(1), 5815. https://doi.org/10.1038/s41467-021-25983-9

Gordon, R. P., Lautz, L. K., Briggs, M. A., & McKenzie, J. M. (2012). Automated calculation of vertical pore-water flux from field temperature time series using the VFLUX method and computer program. Journal of Hydrology, 420, 142-158.

Hatch, C. E., Fisher, A. T., Revenaugh, J. S., Constantz, J., & Ruehl, C. (2006). Quantifying surface water–groundwater interactions using time series analysis of streambed thermal records: Method development. Water Resources Research, 42(10).

Keery, J., Binley, A., Crook, N., & Smith, J. W. (2007). Temporal and spatial variability of groundwater–surface water fluxes: Development and application of an analytical method using temperature time series. Journal of Hydrology, 336(1-2), 1-16.

Knapp, J. L. A., González-Pinzón, R., Drummond, J. D., Larsen, L. G., Cirpka, O. A., & Harvey, J. W. (2017). Tracer-based characterization of hyporheic exchange and benthic biolayers in streams. *Water Resources Research*, *53*(2), 1575-1594. https://doi.org/10.1002/2016WR019393

Krause, S., Blume, T., & Cassidy, N. J. (2012). Investigating patterns and controls of groundwater up-welling in a lowland river by combining Fibre-optic Distributed Temperature Sensing with observations of vertical hydraulic gradients. *Hydrology and Earth System Sciences*, *16*(6), 1775-1792. https://doi.org/10.5194/hess-16-1775-2012

Rivett, M. O., Ellis, P. A., Greswell, R. B., Ward, R. S., Roche, R. S., Cleverly, M. G., Walker, C., Conran, D., Fitzgerald, P. J., Willcox, T., & Dowle, J. (2008). Cost-effective mini drive-point piezometers and multilevel samplers for monitoring the hyporheic zone. *Quaterly Journal of Engineering Geology and Hydrogeology*, *41*, 49-60.

Rühlmann, J., Körschens, M., & Graefe, J. (2006). A new approach to calculate the particle density of soils considering properties of the soil organic matter and the mineral matrix. *Geoderma*, *130*(3-4), 272-283. https://doi.org/10.1016/j.geoderma.2005.01.024

Schaper, J. L., Posselt, M., McCallum, J. L., Banks, E. W., Hoehne, A., Meinikmann, K., Shanafield, M. A., Batelaan, O., & Lewandowski, J. (2018). Hyporheic Exchange Controls Fate of Trace Organic Compounds in an Urban Stream. *Environmental Science & Technology*, *52*(21), 12285-12294. https://doi.org/10.1021/acs.est.8b03117

Seeberg-Elverfeldt, J., Schlüter, M., Feseker, T., & Kölling, M. (2005). Rhizon sampling of porewaters near the sediment-water interface of aquatic systems. Limnology and oceanography: Methods, 3(8), 361-371.

Shotbolt, L. (2010). Pore water sampling from lake and estuary sediments using Rhizon samplers. Journal of Paleolimnology, 44(2), 695-700.

Song, J., Luo, Y., Zhao, Q., & Christie, P. (2003). Novel use of soil moisture samplers for studies on anaerobic ammonium fluxes across lake sediment–water interfaces. Chemosphere, 50(6), 711-715.

---

## Author Response (AR2)

**Author's response**

Answer to Anonymous Referee #1

**Technical Note: Testing the effect of different pumping rates on pore-water sampling for ions, stable isotopes and gas concentrations in the hyporheic zone**

We want to thank the reviewer for his positive assessment and helpful comment.

I have only one small comment: In the caption of figure S9, explain that mu is the mean and sigma is the standard deviation (if that is the case). Moreover, as they refer to dispersivities, they have units (e.g., m), that should be mentioned.

Thank you for this thoughtful note. We have adjusted the caption of Fig. S9 accordingly:

"Monte Carlo analysis for thermal dispersivity. Three scenarios were tested for mean $\mu$ and standard deviation $\sigma$ of the thermal dispersivity parameter $\beta$ in m. Results were generated with n=100 runs for each scenario. Shading indicates 95% confidence intervals for each scenario. The results were calculated with the software package VFLUX and the Hatch amplitude method."